# Systematic Review of Fear of Cancer Recurrence Patient-Reported Outcome Measures: Evaluating Methodological Quality and Measurement Properties Using the COSMIN Checklist

**DOI:** 10.3390/healthcare13172165

**Published:** 2025-08-29

**Authors:** Christine Maheu, Wing Lam Tock, Peter Fisher, Jacqueline Galica, Mina Singh, Isabel Centeno, Maude Hébert, Chelsea Moran, Paulina Pietruczuk, Andreas Dinkel, Caroline Zwaal, Belinda Thewes, Tania Estapé

**Affiliations:** 1Ingram School of Nursing, McGill University, Montréal, QC H3A 2M7, Canada; 2Centre de Recherche du Centre Hospitalier de l’Université de Montréal, Montréal, QC H2X 0A9, Canada; wing.lam.tock@umontreal.ca; 3Department of Primary Care and Mental Health, University of Liverpool, Liverpool L69 7ZX, UK; peter.fisher@liverpool.ac.uk; 4School of Nursing, Queen’s University, Kingston, ON K7L 3N6, Canada; jacqueline.galica@queensu.ca; 5School of Nursing, York University, Toronto, ON M3J 1P3, Canada; minsingh@yorku.ca; 6Cáncer Institute, Hospital Zambrano Hellion, San Pedro Garza García 66260, Mexico; icenteno@tecsalud.mx; 7Département des Sciences Infirmières, Université du Québec à Trois-Rivières, Trois-Rivières, QC G8Z 4M, Canada; maude.hebert@uqtr.ca; 8Department of Cardiac Prevention and Rehabilitation, University of Ottawa Heart Institute, Ottawa, ON K1Y 4W7, Canada; chmoran@ottawaheart.ca; 9Department of Nursing, Jewish General Hospital, Montreal, QC H3T 1E2, Canada; paulina.pietruczuk@gmail.com; 10Department of Psychosomatic Medicine and Psychotherapy, TUM University Hospital, School of Medicine and Health, Technical University of Munich, 81675 Munich, Germany; a.dinkel@tum.de; 11Program in Evidence-Based Care, McMaster University, Hamilton, ON L8V 1C3, Canada; zwaalc@mcmaster.ca; 12The Health Psychology Clinic, Moruya, NSW 2537, Australia; belinda@healthpsychclinic.com.au; 13Psychosocial Oncology Department, FEFOC Fundación, 08021 Barcelona, Spain; testape@fefoc.org

**Keywords:** fear of cancer recurrence, patient-reported outcomes, measurement properties, COSMIN, survivorship, screening tools

## Abstract

**Background:** Fear of cancer recurrence (FCR) is a common, distressing concern among cancer survivors, with implications for psychological well-being and quality of life. Despite the proliferation of patient-reported outcome measures (PROMs) to assess FCR, consensus on the most valid and reliable instruments remains limited. **Objective:** To conduct a comprehensive psychometric evaluation of FCR PROMs developed or validated between 2011 and 2023, using the full COSMIN methodology. This review builds on earlier work by systematically appraising both newly developed and adapted instruments to inform evidence-based PROM selection for clinical and research use. **Methods:** This review followed the COSMIN 10-step procedure and PRISMA 2020 guidelines. Six databases were searched from January 2011 to December 2023. A search cut-off of 31 December 2023 was applied to align with COSMIN feasibility recommendations and ensure sufficient time and resources for full psychometric evaluation across all included PROMs. Eligible studies included adults diagnosed with cancer and reported on the psychometric evaluation of a self-reported, Likert-scale-based FCR PROM. PROMs were appraised across eight psychometric properties using COSMIN’s Risk of Bias checklist, criteria for good measurement properties, and modified Grading of Recommendations Assessment, Development, and Evaluation (GRADE) for evidence quality. Instruments were categorized as A, B, or C according to COSMIN’s recommendations. **Results:** Of the 34 PROMs evaluated across 32 studies, 28 achieved COSMIN Category A status, indicating sufficient psychometric quality for clinical or research use. Among the 19 FCRI and FoP-Q instruments validated in new cultural or population contexts, 15 received a Category A rating, reflecting strong cross-cultural performance. Short forms such as the FCRI-SF and FoP-Q-SF demonstrated consistent psychometric strength across French, English, Portuguese, and Asian-language versions. Of the 14 newly developed PROMs—including the CWS-6, FCR4/7, CARQ-4, and FCR-1—12 achieved Category A. The FCR-1 was the only single-item PROM to demonstrate responsiveness, supporting its use in both screening and longitudinal monitoring. For practical guidance, five PROMs (FCR-1, FCRI-SF, FoP-Q-SF, FCR-7, and CWS) emerged as the most strongly supported for clinical use. **Conclusions:** This review provides a comprehensive evaluation of contemporary FCR PROMs and highlights substantial progress in the development of valid, reliable, and culturally adaptable tools. The FCRI, FoP-Q-SF, and several newly developed short forms, such as FCR7, CWS, and particularly, the FCR-1, emerge as strong candidates for use in clinical screening and longitudinal assessment. These findings offer clear evidence-based guidance for instrument selection in research and survivorship care. **PROSPERO registration:** CRD42023453783.

## 1. Introduction

Fear of cancer recurrence (FCR) remains one of the most common and distressing unmet needs among cancer survivors [1,2,3]. A systematic review of 130 studies estimated that approximately 49% of cancer survivors experience moderate to high levels of FCR, with 7% reporting high to severe levels [4]. More recently, Luigjes-Huizer et al. 2022 [1] conducted a meta-analysis focused specifically on clinical levels of FCR. Among 11,226 patients and survivors, 58.8% scored ≥ 13 on the Fear of Cancer Recurrence Inventory—Short Form (FCRI-SF) [5], a widely used and highly cited FCR clinical screening measure [6]. Additionally, 45.1% scored ≥ 16, and 19.2% scored ≥ 22, reinforcing the high prevalence of clinically significant FCR and the urgent need for validated Patient Reported Outcome Measures (PROMs) suitable for use in both routine clinical care and research [7].

Identifying robust PROMs to assess FCR remains a recognized clinical and research priority [8]. However, no consensus has emerged regarding which measures are most appropriate for routine screening or longitudinal assessment [9]. In 2012, Thewes et al. [6] published a foundational systematic review of FCR measures available up to 2010, using the Medical Outcomes Trust criteria to appraise psychometric properties. That review identified 20 relevant multi-item measures, many with evidence supporting their reliability and validity, but noted that few brief measures (2–10 items) had undergone comprehensive validation.

Since then, substantial progress has occurred in both the conceptualization and measurement of FCR. New measures have been developed, and updated consensus definitions, including a Delphi process defining the clinical (pathological) form, have emerged [8,10]. FCR is now defined as “the fear, worry, or concern that the cancer will return or progress” [8], with high FCR characterized by persistent worry, intrusive thoughts, avoidance behaviors, and functional impairments [4,6,8]. High FCR is associated with psychological distress, post-traumatic symptoms, impairments in functioning [2,11], and reduced quality of life [12,13,14].

In parallel, the international psycho-oncology community has mobilized around FCR measurement. In 2015, the forwards special interest group of the International Psycho-Oncology Society (IPOS) was established to address global priorities related to FCR, including PROM standardization. With over 45 members across 11 countries, the group has identified the development and appraisal of valid FCR measures as a key research priority.

Previous systematic reviews provided important early syntheses but did not apply the full COSMIN methodology or integrate formal quality rating using GRADE. They also did not incorporate single-item screening tools or systematically evaluate cross-cultural adaptations published since 2011. This current review therefore offers the first comprehensive, COSMIN-based appraisal of both multi-item and single-item PROMs, together with a structured comparison of translated and culturally validated versions.

What This Study Adds: This review advances the literature by

Applying standardized COSMIN ratings across eight psychometric domains;Using the modified GRADE approach to assess quality of evidence;Categorizing PROMs according to COSMIN’s A/B/C recommendations;Emphasizing the link between content validity, structural validity, and internal consistency;Synthesizing findings across translated and adapted versions of each instrument;Providing context-specific recommendations for clinical and research tool selection.

The primary research question guiding this review was the following: which PROMs assessing FCR in adults with a confirmed cancer diagnosis demonstrate sufficient measurement properties, as evaluated using the COSMIN criteria?

## 2. Method

### 2.1. Framework for Conducting the Systematic Review

This systematic review was conducted in accordance with the COSMIN methodology for systematic reviews of PROMs [15,16,17] and reported following PRISMA 2020 guidelines for systematic reviews [18]. The full ten-step COSMIN procedure guided each phase of the review (see Figure 1) and included the following: formulating the aim, defining eligibility criteria, developing and executing the search strategy, selecting abstracts and full texts, extracting data, evaluating each measurement property, appraising the quality of evidence, and drawing final recommendations based on evidence synthesis [15]. The following subsections describe how each of the ten COSMIN steps was applied in this review.

#### 2.1.1. Part A Steps 1-4: Perform the Literature Search 

Step 1—Formulate Aim: The aim was to identify and evaluate new FCR PROMs and newly validated applications of existing FCR instruments published since the 2012 systematic review by Thewes et al. [6], focusing on studies published between January 2011 and December 2023. PROMs were included if they underwent psychometric evaluation according to COSMIN standards. As well, included studies had to either introduce a new self-report measure for FCR or evaluate the performance of an existing validated tool in a new population, language, or cultural context, accompanied by a psychometric evaluation. Only studies reporting empirical data on at least one COSMIN-defined measurement property were retained.

Step 2—Eligibility Criteria: Eligibility criteria were defined using the PICO framework. Studies were eligible if they included adults (≥18 years) with a confirmed diagnosis of malignant cancer. Studies were excluded if fewer than 80% of the participants had a confirmed cancer diagnosis, or if the population consisted of individuals at genetic risk (e.g., BRCA1/2 mutation carriers), individuals with benign tumors (e.g., DCIS, BPH), or participants from general cancer screening populations (e.g., mammography attendees). Pediatric and AYA populations were also excluded, as the review focused on PROMs validated in adult cancer populations with a confirmed diagnosis. Only English language published quantitative studies were considered if they reported the psychometric evaluation of a self-reported, Likert-scale-based FCR PROM and documented research ethics approval or informed consent procedures. Eligible studies were required to report on at least one COSMIN-defined measurement property. Grey literature (e.g., theses, non-peer-reviewed sources) was excluded to maintain methodological rigor, consistent with COSMIN recommendations. Although no formal statistical assessment of publication bias was conducted due to the heterogeneity of psychometric outcomes, prospective PROSPERO registration and transparent eligibility criteria reduced risk of bias in study selection. Eligibility was subsequently refined to also encompass single-item FCR measures, reflecting their increasing use as rapid screening tools.

Step 3—Search Strategy: An initial comprehensive search was conducted by the institutional librarian (F.F) of the first author, covering July 2011 to August 2019 across five databases (MEDLINE, Embase, CINAHL, PsycINFO, AMED) and through hand-searching of reference lists. This search yielded 6524 records, which were uploaded to Covidence systematic review software (Veritas Health Innovation, Melbourne, Australia; www.covidence.org), for duplicate removal and screening. After title/abstract and full-text review, 16 PROMs from 16 records were retained. Three update searches were subsequently conducted. The first update, from September 2019 to July 2020, identified 1464 records, of which 4 PROMs were retained from 4 records. The second update, covering August 2020 to November 2022, yielded 426 records, resulting in 6 PROMs retained from 5 records. The final update, from December 2022 to December 2023, identified 1067 records, of which 8 PROMs were retained from 7 records. In total, 32 records describing 34 distinct PROMs were retained for full COSMIN evaluation. Full search terms are provided in Table 1, and the selection process is summarized in the PRISMA flow diagram (Figure 2).

Step 4—Abstracts and full-text selection: Selection of abstracts and full-text articles was performed by four teams of two reviewers per pair. Screening was managed using Covidence systematic review software, which supported identification of duplicates, abstract/full-text screening, and data extraction. If a study appeared potentially eligible to at least one reviewer, full text was retrieved. Any disagreements between reviewers were resolved through discussion, and if unresolved, adjudicated by a third reviewer (C.M., W.L.T., or T.E.). Reference lists of included articles were also reviewed for additional eligible studies, consistent with COSMIN’s recommendation to ensure comprehensiveness.

#### 2.1.2. Part B Steps 5–7: Evaluating the Measurement Properties of the Included PROMs

For part B, four teams of two reviewers per pair independently extracted key study characteristics and psychometric data from assigned PROMs into structured tables. Extracted domains included the following: instrument purpose, population, language, subscale structure, content validity, and all eight COSMIN measurement properties. Following extraction, 50% of the PROMs were double-checked by two reviewers (C.M., M.S.) to verify agreement and discuss any discrepancies, most of which involved clarifying the population type or correcting misidentified countries rather than language. Once data extraction for all 34 PROMs was complete, two co-authors (C.M., W.L.T.) independently verified all entries to ensure completeness and accuracy. As per COSMIN guidelines, measurement properties were evaluated independently without weighting. However, content validity, as well as the combination of structural validity with internal consistency, were prioritized in determining overall instrument quality and category placement.

Step 5—Content Validity: Content validity was assessed by evaluating whether the PROM was developed using qualitative methods such as concept elicitation interviews, expert feedback, and cognitive interviews. PROMs that demonstrated conceptually aligned item development were rated as having sufficient content validity. For short forms derived from well-established tools (e.g., FCRI-SF), content validity was inferred to be sufficient if item selection preserved the original conceptual domains and item wording remained unchanged. In addition, consistent with COSMIN’s broader recommendations for PROM adaptation and use in new populations, each of the 34 included PROMs was reviewed to determine whether the study authors had (1) obtained permission to translate or adapt the original instrument, (2) involved the original instrument developer as part of the study team, and (3) documented their approach to maintaining conceptual and linguistic equivalence. While COSMIN does not explicitly require permission from the original developer, it endorses international standards (e.g., ISPOR, Beaton, WHO) that treat author contact as a best practice to ensure conceptual fidelity. These data were systematically recorded and are reported in Appendix A to support transparency, ethical adaptation practices, and interpretation of content validity, particularly in translated or modified versions.

Steps 6 to 7: Internal Structure and Psychometric Evaluation: Each PROM was evaluated across eight psychometric properties listed in Table 2, following COSMIN’s standardized approach for systematic reviews of outcome measurement instruments. These measurement properties (mp) were as follows: (mp1) structural validity, (mp2) internal consistency, (mp3) reliability, (mp4) measurement error, (mp5) criterion validity, (mp6) hypothesis testing for construct validity, (mp7) cross-cultural validity, and (mp8) responsiveness.

The methodological quality of each study related to a given property was rated using the COSMIN four-point scale—very good, adequate, doubtful, or inadequate—with the “worst score counts” principle applied to determine the overall score within each property domain. This ensured that any methodological weakness within a domain would appropriately influence the confidence in the property’s findings.

Each psychometric property was subsequently compared and rated for sufficiency using COSMIN’s predefined criteria for good measurement properties, as detailed in Table 2. Ratings included the following: “+” (sufficient) when criteria for adequacy were met, “−” (insufficient) when the results fell below acceptable thresholds, and “?” (indeterminate) when data were incomplete or did not allow for a confident judgment. Examples of COSMIN criteria for good measurement properties are summarized in Table 2. For instance, structural validity was considered sufficient when unidimensionality was supported by CFA with acceptable fit indices (e.g., CFI or TLI ≥ 0.95; RMSEA ≤ 0.06), and reliability was accepted when Cronbach’s α or ICC was ≥0.70. When thresholds were not met or evidence was unclear, properties were rated as indeterminate.

The rating approach and thresholds used ensured consistency and rigor in comparing PROMs across studies, in accordance with COSMIN guidance [15,16,17].

Evaluating the Quality of the PROMs: Evaluation of PROM quality followed the COSMIN guidelines for systematically reviewing PROMs [15,16,17]. Each PROM was independently assessed across eight psychometric properties using COSMIN’s Risk of Bias checklist to determine methodological quality [17]. Ratings for each property were assigned based on COSMIN’s criteria for good measurement properties. The overall quality of the evidence supporting each rating was then graded using the modified GRADE approach, which considers risk of bias, inconsistency, imprecision, and indirectness. Based on these evaluations, each PROM was categorized into COSMIN Category A, B, or C, providing a structured summary of its psychometric adequacy for clinical or research use. This approach ensured a transparent, standardized, and reproducible process across all instruments included in the review.

Grading considered four potential downgrading factors: risk of bias, inconsistency, imprecision, and indirectness. Each psychometric property was assigned a level of evidence—high, moderate, low, or very low—to reflect confidence in the observed measurement performance. Based on the sufficiency of each property and the quality of supporting evidence, PROMs were then assigned to one of three COSMIN recommendation categories. This classification provides a practical framework for determining whether an instrument is suitable for use, requires further validation, or should be avoided.

Category A: PROMs with sufficient content validity (either demonstrated or inferred from original development) and sufficient internal consistency, based on at least low-quality evidence of sufficient structural validity. As per COSMIN, internal consistency can only be interpreted when unidimensionality is established in the population under study. These PROMs are considered suitable for use.

Category B: PROMs with potential utility but requiring further validation. These instruments had at least one indeterminate or insufficient rating, or the evidence was of low or very low quality, but no fatal flaws were identified. Such PROMs may be cautiously used in exploratory settings.

Category C: PROMs for which there was high-quality evidence of at least one insufficient measurement property. These instruments are not recommended for use in their current form.

All PROMs were assessed for risk of bias using procedures outlined in the COSMIN Risk of Bias checklist. Two independent review teams (C.M. and W.L.T.; M.S. and T.E.) evaluated all measurement properties of each PROM. Each team first completed independent assessments, followed by cross-review of the other team’s evaluations. Discrepancies were resolved through discussion with a third reviewer who had not been involved in the initial assessment of that PROM. No automation tools were used in this process. The COSMIN Risk of Bias tool to assess the quality of studies on reliability and measurement error of outcome measurement instruments (User Manual, Version 1.0, January 2021) guided all evaluations.

Special considerations applied to certain PROM types. For instruments tested in new populations or languages, COSMIN requires re-evaluation of both structural validity and internal consistency in the new context.

Translated PROMs could only qualify as Category A if sufficient evidence supported both properties. For single-item tools (e.g., the FCR-1), structural validity and internal consistency were marked “Not Applicable”, since these domains require multiple items. Evaluation instead focused on remaining applicable properties.

Cross-cultural validity was rated sufficient only when formal statistical testing—such as DIF analysis or multi-group CFA (MG-CFA)—was conducted. When such analyses were absent, even if linguistic translation and pilot testing were reported, cross-cultural validity was rated indeterminate (“?”) and methodological quality downgraded to “inadequate”, in accordance with COSMIN guidance.

Regarding construct validity (hypothesis testing), sufficient ratings (“+”) were assigned when statistically significant results aligned with predefined hypotheses in terms of direction and magnitude. Comparator tools such as the IES, HADS, and GAD-7 were accepted as measuring conceptually related constructs (e.g., anxiety, depression, cancer-related distress). When hypotheses were clearly defined, comparator selection was justified, and COSMIN thresholds for sample size were met (n ≥ 50 per hypothesis), methodological quality was rated “very good” and the quality of evidence as “high.”

For criterion validity, in the absence of a true gold standard for FCR, COSMIN allows comparators that measure closely related constructs, such as the FCRI or FCRI-SF. PROMs were rated as sufficient (“+”) when such conceptually aligned comparators were used. When instruments assessed unrelated domains (e.g., general health, stress, somatization), the property was rated indeterminate (“?”) and methodological quality downgraded to “doubtful” or “inadequate,” depending on justification and analytic approach.

For structural validity and reliability for screening variants of established PROMs derived from previously validated outcome measures (e.g., the Severity subscale of the FCRI [5]), structural validity was assessed based on existing factor analytic evidence. If the literature demonstrated that the original subscale had sufficient unidimensionality and model fit indices consistent with COSMIN standards (e.g., CFI > 0.95, RMSEA < 0.06, or comparable EFA support), and there were no serious flaws in the analytic approach, the structural validity of the derived short form could be rated as sufficient. When a subscale is used independently in a new study, COSMIN requires structural validity to be reassessed, or at minimum explicitly justified. As no CFA was repeated in these screening validations, the methodological quality was conservatively rated as Adequate, the quality of evidence as Moderate, and the overall property rating as sufficient (+). This approach was applied only when the short-form PROM was used for screening purposes, was derived from a subscale with established unidimensional structure, and was validated either in the same language or as part of a cross-cultural adaptation study. See Appendix A for detailed PROM-level ratings and methodological justifications.

#### 2.1.3. Part C Steps 8–10: Selecting a PROM

Finally, COSMIN final steps 8 to 10 included drawing conclusions on the interpretability and feasibility of the PROMs (step 8), formulating recommendations of their use based on all evidence (step 9), and reporting the systematic review (step 10), as completed in the results section.

## 3. Results

Presentation of the 34 Distinct FCR PROMs from 32 Reports.

A total of 32 records were retained, representing 34 distinct PROMs. Two records (Xu et al. [19] and Bergerot et al. [20]) each evaluated more than one PROM, resulting in 34 PROM evaluations included in this review. Across these 34 PROMs, 28 measures achieved Category A, five were classified as Category B, and one as Category C. Descriptive characteristics of the included PROMs are provided in Table 3.

In total, 267 measurement property ratings were extracted. Of these, 48% were rated sufficient (+), 2% insufficient (−), and 50% indeterminate (?). These determinations incorporate both the COSMIN-based property evaluations and their associated quality of evidence ratings, as summarized in Table 4, and detailed in Appendix A.

For clarity and consistency, the PROMs are organized into three interpretive categories based on their development and validation context: established PROMs with cultural or population-specific validations; short forms or psychometric adaptations of existing instruments; and newly developed PROMs introduced after the 2012 FCR PROMs systematic review by Thewes et al. [6].

### 3.1. Established PROMs with Cultural or Population-Specific Validation

This category includes established FCR PROMs that were previously identified in the systematic review by Thewes et al. [6] and have since been validated in new populations, languages, or healthcare contexts. These updated validations provide important evidence of cultural adaptation, structural stability, and psychometric performance across diverse settings, assessed using the updated COSMIN methodology [15,17].

(a)Fear of Cancer Recurrence Inventory (FCRI) [5]—Adaptation and Validation in New Languages and Populations

This subsection reviews studies that adapted and validated the long-form FCRI [5], originally developed in French, for use in diverse language and cultural contexts. Six studies were identified, involving English-, Mandarin-, Korean-, Dutch-, Danish-, and Chinese-speaking cancer survivor populations. These adaptations followed standard translation procedures and varied in psychometric scope and rigor. Each version is evaluated independently below. See Table 4 and Appendix A for their full psychometric profiles.

Lebel et al.—FCRI English Version [21]: Lebel et al. translated and validated the long-form English version of the FCRI in a sample of 350 Canadian cancer survivors. The translation process followed a forward–backward method with pilot testing to ensure semantic and cultural equivalence. CFA supported the original seven-factor structure, with excellent model fit (CFI = 0.98; RMSEA = 0.06). Internal consistency was excellent for the total scale (α = 0.96) and strong across all subscales (α = 0.71–0.94). Test–retest reliability was high (ICC = 0.94), and construct validity was confirmed via hypothesis-driven correlations with the FACT-G fear item (r = 0.68) and the EORTC emotional functioning subscale (r = −0.47). Criterion validity was rated sufficient based on moderate correlations with the EORTC emotional subscale, while cross-cultural validity was explored in a bilingual subgroup, revealing no significant differences between the English and French versions; however, no formal DIF or invariance testing was performed. Measurement error and responsiveness were not evaluated. According to COSMIN criteria, the English FCRI meets requirements for Category A, supported by high-quality evidence for structural validity, internal consistency, reliability, and construct and criterion validity. It is recommended for use in clinical and research contexts across English-speaking cancer populations.

Shin et al.—FCRI Korean Version [22]: Shin et al. adapted and validated the Korean version of the FCRI (K-FCRI) in a sample of 444 Korean cancer survivors. Translation followed a forward–backward process with expert panel review to ensure cultural and linguistic equivalence. CFA supported the original seven-factor structure, with model fit indices meeting COSMIN thresholds (CFI = 0.90; RMSEA = 0.06). Internal consistency was excellent across the total scale and subscales (Cronbach’s α = 0.85–0.91), and test–retest reliability exceeded the acceptable threshold across domains (ICC > 0.70), supporting temporal stability. Construct validity was confirmed through strong correlations with depression (r = 0.76), anxiety (r = 0.72), and psychological distress (r = 0.66). Criterion validity was supported by ROC analysis using a clinical anchor, yielding an AUC of 0.77 for distinguishing clinically significant FCR. Cross-cultural validity and measurement error were not assessed. According to COSMIN guidelines, the K-FCRI met criteria for sufficient structural validity, internal consistency, test–retest reliability, construct and criterion validity, and was rated Category A. It is recommended for clinical and research use in Korean-speaking oncology populations.

van Helmondt et al.—FCRI Dutch Version (FCRI-NL) [23]: Van Helmondt et al. validated the Dutch version of the long-form FCRI (FCRI-NL) in a sample of 255 cancer survivors. Translation followed a forward–backward process with cultural adaptation and pilot testing. CFA supported the original seven-factor structure, with acceptable model fit (AGFI = 0.93; NFI = 0.93; SRMR = 0.08). Internal consistency was high across the total scale (α = 0.93) and all subscales (α = 0.75–0.92). Test–retest reliability was strong, with ICC = 0.84 for the total score and subscale ICCs ranging from 0.56 to 0.87, supporting temporal stability. Construct validity was confirmed through significant correlations with related constructs such as anxiety (STAI, r = 0.63), and divergent validity was supported by weaker associations with unrelated constructs (e.g., extraversion, r = −0.20). Criterion validity was supported via correlations with theoretically aligned measures. Cross-cultural validity, responsiveness, and measurement error were not evaluated. Nonetheless, based on sufficient structural validity, internal consistency, test–retest reliability, and construct validity, the FCRI-NL was rated Category A under COSMIN and is recommended for research and clinical use in Dutch-speaking cancer populations.

Hovdenak Jakobsen et al.—FCRI-Danish [24]: Hovdenak Jakobsen et al. conducted an initial validation of the Danish version of the FCRI in a sample of 69 colorectal cancer survivors (mean age = 67.3 years; 56.5% male). Translation followed a forward–backward method using the English version of the FCRI [21], with additional pilot testing in gynecological cancer survivors. Content validity was supported through expert review and cognitive interviews. Structural validity was not assessed, as the original seven-factor model was assumed but not empirically tested. Internal consistency could not be rated due to the absence of factor analysis and Cronbach’s alpha reporting. Test–retest reliability was strong (ICC = 0.84), and construct validity was supported by moderate correlations with cancer-related worry (r = 0.49) and age (r = −0.29). Responsiveness was demonstrated through significant pre–post scan score reductions (mean difference = 4.9, *p* = 0.005). No data were reported on measurement error, criterion validity, or cross-cultural validity. Based on COSMIN guidelines, the Danish FCRI demonstrated sufficient reliability, responsiveness, and construct validity, but the lack of structural validation limits its interpretability. It was rated Category B, indicating that further validation is needed before it can be recommended for routine clinical use or longitudinal monitoring.

Liu et al.—FCRI (English and Mandarin Versions) [25]: Liu et al. validated the English and Mandarin versions of the long-form FCRI in a diverse sample of 219 Singaporean cancer survivors (Mandarin: *n* = 109; English: *n* = 110). The translation process followed standard forward–backward procedures, and both language versions were evaluated for equivalence. CFA supported the original seven-factor structure, with acceptable fit indices (CFI = 0.91; RMSEA = 0.06; SRMR = 0.08). Internal consistency was excellent for both versions (α = 0.95 for English; α = 0.93 for Mandarin), and test–retest reliability was high (ICC = 0.92 and 0.86, respectively). Construct validity was supported by significant correlations with anxiety, fear of progression, and quality of life. Cross-cultural validity was formally assessed using an MIMIC model, which confirmed measurement invariance between the two versions. Criterion validity was evaluated using the Fear of Recurrence Questionnaire by Northouse [53], with strong convergent correlations (r = 0.61). Although measurement error and responsiveness were not assessed, the FCRI demonstrated sufficient structural validity, internal consistency, reliability, and cross-cultural equivalence. It was rated Category A under COSMIN and is recommended for clinical and research use in English- and Mandarin-speaking populations.

Xu et al.—FCRI-C (Long Form) [19]: Xu et al. adapted and validated the full Chinese version of the FCRI (FCRI-C) in a sample of 326 lymphoma survivors. Translation followed a standardized forward–backward process with expert review. CFA supported a bifactor structure with acceptable model fit (CFI = 0.920; RMSEA = 0.063). Internal consistency was excellent across the total scale (α = 0.95) and subscales (α = 0.78–0.97). Test–retest reliability over a one-week interval was high (ICC = 0.82), indicating temporal stability. Criterion validity was evaluated using the PHQ-9, yielding an AUC of 0.77, which meets COSMIN standards for a conceptually related comparator. However, cross-cultural validity was rated insufficient due to moderate DIF across sex and age groups. Measurement error and responsiveness were not evaluated. Based on COSMIN criteria, the FCRI-C long form meets requirements for Category A, supported by sufficient evidence for structural validity, internal consistency, reliability, and criterion validity. It is recommended for use in clinical and research settings involving monolingual Chinese cancer populations, though additional validation is needed to confirm cross-cultural equivalence and longitudinal responsiveness.

(b)Concerns About Recurrence Scale (CARS) [54]—Adaptation and Validation in a New Language and Population

This subsection reviews the Japanese adaptation of the original CARS [54], developed by Momino et al. [26]. The CARS-J was culturally adapted and evaluated in a sample of Japanese breast cancer survivors and is the only identified version validated outside the original English context. See Table 4 and Appendix A for its full psychometric profile.

Momino et al.—CARS-J [26]: One adaptation of the original CARS [54] was identified: the Japanese version (CARS-J) developed and evaluated by Momino et al. [26]. The 29-item multidimensional PROM was translated and culturally adapted using a forward–backward translation procedure and tested in 245 Japanese breast cancer survivors. Exploratory factor analysis supported a revised factor structure distinct from the original, with the merging of Health and Death Worries and the emergence of a new Self-Valued Worries domain. Structural validity was rated sufficient based on exploratory factor analysis explaining 59.2% of the total variance with item loadings > 0.40. Internal consistency was strong (α = 0.86–0.94). Cross-cultural validity was rated as indeterminate due to the absence of DIF or invariance testing. Construct and criterion validity were both rated sufficient, supported by significant correlations with HADS anxiety and depression subscales. Responsiveness and test–retest reliability were not assessed. The CARS-J received a COSMIN Category A rating, based on sufficient evidence for content validity, structural validity, internal consistency, and construct and criterion validity.

### 3.2. FCRI-Derived Short Forms—Development, Validation, and Cross-Cultural Adaptation

This section presents revised and shortened versions of existing FCR PROMs, developed to reduce response burden and improve clinical applicability. All instruments in this section are derived from the original 42-item FCRI by Simard and Savard [5]. The following subsections group studies into (a) short forms developed through IRT modeling and cross-cultural validation, (b) alternative short versions focused on item reduction strategies, and (c) culturally adapted shorter forms validated in non-English populations. See Table 4 and Appendix A for their full psychometric profiles.

(a)FCRI [5]—Shortened and Adapted Versions

This subsection reviews three shortened versions of the original 42-item FCRI [5], developed to enhance feasibility and reduce respondent burden while preserving key FCR dimensions. These adaptations were tested in Australian, Turkish, and Chinese cancer populations using either item response theory or expert-guided item reduction. Each tool is evaluated individually below.

Costa et al.—16-item FCRI Short Form [27]: Costa et al. used item response theory (IRT) modeling to derive a 16-item short form of the 42-item FCRI, using data from 286 Australian melanoma survivors. Items were selected across all seven subscales to preserve conceptual breadth and optimize discrimination based on IRT performance. CFA supported the original seven-factor structure of the full FCRI (RMSEA = 0.051; CFI = 0.956; TLI = 0.953), but the short form was not independently evaluated for structural validity. Content validity was inferred based on preservation of original item wording and subscale distribution, though no patient involvement or cognitive debriefing was conducted. Internal consistency, test–retest reliability, criterion validity, and responsiveness were not assessed for the short form, and no Cronbach’s alpha was reported. As such, only structural validity of the parent scale was confirmed. According to COSMIN criteria, the 16-item version qualifies as a promising instrument but lacks the independent psychometric evaluation required for full endorsement. It was rated Category B and may be cautiously used in research or clinical contexts focused on reducing respondent burden, with further validation strongly recommended.

Eyrenci and Sertel Berk—Turkish FCRI [28]: Eyrenci and Sertel Berk validated a 24-item Turkish adaptation of the FCRI in a sample of 375 cancer survivors. Translation followed a forward–backward process and was approved by the original developers. CFA supported a five-factor structure, showing good model fit (RMSEA = 0.067; CFI = 0.92), and the total scale demonstrated excellent internal consistency (α = 0.94). Construct validity was supported through moderate to strong correlations with depression (PHQ-9, r = 0.47), anxiety (GAD-7, r = 0.62), and trauma-related symptoms (IES-R Intrusion, r = 0.70). Cross-cultural validity could not be determined, as no item-level DIF or measurement invariance testing was conducted. Similarly, measurement error, test–retest reliability, and responsiveness were not evaluated. Despite these limitations, the adapted Turkish FCRI met COSMIN criteria for sufficient content validity, structural validity, internal consistency, and construct validity. It was classified as a Category A instrument and is recommended for clinical and research use with Turkish-speaking populations. Further validation is needed to confirm its temporal stability and cross-group equivalence.

Xu et al.—FCRI-C Short Form [19]: Xu et al. concurrently validated the full FCRI-C and developed a 10-item short form to enhance clinical feasibility. Items were selected using item response theory and factor loading criteria from four subscales—Severity, Psychological Distress, Functioning Impairments, and Insight—ensuring broad conceptual coverage beyond the original Severity subscale used in the FCRI-SF. CFA supported a unidimensional structure with acceptable fit (RMSEA = 0.066; CFI = 0.912), and internal consistency was excellent (α = 0.95). Test–retest reliability was high (ICC = 0.87). Criterion validity, assessed against the PHQ-9, yielded an AUC of 0.80, meeting COSMIN thresholds for a conceptually related comparator. However, cross-cultural validity was rated insufficient due to moderate DIF across age and gender groups. Measurement error and responsiveness were not evaluated. According to COSMIN criteria, the short form meets requirements for Category A, with sufficient evidence for content validity, structural validity, internal consistency, and reliability. It is recommended for use in monolingual Chinese clinical settings, but further work is needed to confirm its cross-cultural and longitudinal applicability. Like Costa et al. (2016) [27], who emphasized that their 16-item version complements rather than replaces the FCRI-SF, the 10-item FCRI-C short form should be viewed as a clinically pragmatic alternative with broader conceptual scope.

(b)FCRI Short Form (FCRI-SF) [5]—Screening Variants and Cut-Off Versions

This subsection reviews studies that validated the FCRI-SF, a nine-item standalone screening tool derived from the Severity subscale of the full 42-item FCRI [5]. Four studies assessed the FCRI-SF’s screening utility across French [29], English [30], Chinese [31], and Brazilian Portuguese [20] populations. Each study treated the FCRI-SF as an independent PROM and was evaluated accordingly. See Table 4 and Appendix A for their full psychometric profiles.

Simard and Savard—FCRI-SF (French) [29]: Simard and Savard validated the French version of the FCRI-Short Form (FCRI-SF), a nine-item standalone tool derived from the Severity subscale of the original FCRI. The study was conducted among 60 French-Canadian cancer survivors to assess the tool’s screening performance. Although CFA was not repeated, the original 2009 validation provided sufficient evidence of unidimensionality, with item loadings ranging from 0.45 to 0.84 and an eigenvalue of 4.9, explaining 11.7% of the variance [5]. According to COSMIN, when an established subscale with documented structural validity is used in the same language and population context, the structural validity rating may be justifiably retained. Accordingly, structural validity was rated sufficient. Internal consistency (α = 0.89) was also inferred from the original validation. Construct validity was supported by significant known-group differences and large effect sizes for HADS-Anxiety (d = 0.86) and HADS-Depression (d = 0.56). Criterion validity was confirmed using ROC analysis against the SIFCR structured interview (AUC = 0.88; sensitivity = 88%, specificity = 75%). Content validity was assumed sufficient given conceptual consistency and language fidelity. Based on these properties, the FCRI-SF was rated COSMIN Category A and is recommended for use in clinical and research screening of French-speaking cancer survivors.

Fardell et al.—FCRI-SF (English) [30]: Fardell et al. validated the English version of the FCRI-SF as a clinical screening tool in a large cancer survivor sample across Australia and Canada. The FCRI-SF is a nine-item short form derived from the Severity subscale of the original 42-item FCRI [5]. Although structural validity was not re-tested in this study, it was supported by prior CFA findings from Lebel et al. [21], who validated the English version of the full FCRI including the Severity subscale. Internal consistency (α = 0.89) was cited from Simard and Savard [5], as the item set remained unchanged. The study demonstrated strong criterion validity through ROC analyses using structured clinical interviews, with an AUC of 0.73 in the Australian sample and 0.96 in the Canadian sample. Construct validity was supported by significant correlations with the full FCRI (r = 0.84) and clinician-rated severity scores. Content validity was not explicitly re-evaluated but is presumed adequate given the use of an unchanged, previously validated item set. Test–retest reliability, cross-cultural validity, measurement error, and responsiveness were not assessed. Based on COSMIN criteria, the FCRI-SF received a Category A rating and is recommended for use as a screening tool for FCR in English-speaking cancer survivor populations.

Peng et al.—FCRI-SF (Chinese) [31]: Peng et al. evaluated the Chinese version of the nine-item FCRI-SF in a sample of 207 breast cancer survivors. While the study aimed to assess the scale’s screening capacity for high FCR, it did not include any structural validity testing, no exploratory or CFA was conducted to evaluate the dimensionality of the scale in the Chinese context, which is required by COSMIN for all cross-cultural adaptations. The reported Cronbach’s alpha (α = 0.912) was cited from the original French-language validation by Simard and Savard [5] and was not recalculated using the study’s own sample. Criterion validity was supported through ROC analysis using the HADS-Anxiety subscale as a comparator, yielding an AUC = 0.83, with excellent sensitivity (98.6%) but low specificity (35%) at a cut-off score of ≥ 12. Although the authors referenced a prior validation by another group of researchers in 2018, this source is not accessible for verification and was not adequately described. As such, content validity cannot be confirmed from this article alone. Construct validity, test–retest reliability, measurement error, cross-cultural validity, and responsiveness were not assessed. Based on COSMIN criteria, the Chinese FCRI-SF was rated Category C and is not recommended for use in clinical or research settings until a full validation study is conducted in the target population.

Bergerot et al.—FCRI-SF (Portuguese) [20]: translated and validated the FCRI-SF for use in Brazilian Portuguese among 200 cancer survivors with localized breast cancer or any type of metastatic disease. The cross-cultural adaptation process included forward–backward translation, expert panel review, cognitive debriefing, and pilot testing. Structural validity was supported through CFA, with fit indices meeting COSMIN thresholds (CFI = 0.980; RMSEA = 0.061; SRMR = 0.037), and internal consistency was high (α = 0.887). While the study did not assess test–retest reliability, measurement error, responsiveness, or conduct hypothesis testing or ROC analyses, it met the minimum requirements for COSMIN Category A. This version is recommended for use in Brazilian Portuguese-speaking cancer populations for cross-sectional screening purposes but requires further validation before being used to monitor change over time.

(c)Fear of Progression Questionnaire Short Form (FoP-Q-SF) [32] and FoP-Q [33]—Adaptations and Validations in New Languages and Populations

This subsection reviews six studies that adapted or shortened the original 43-item Fear of Progression Questionnaire (FoP-Q) to improve clinical feasibility while retaining core fear of progression constructs. Versions included translated short forms, rapid screeners, and culturally adapted full-length versions. Studies were conducted in Singapore, Malaysia, Germany, Hong Kong, Portugal, and Iran across a range of cancer populations. Each tool is evaluated individually below. See Table 4 and Appendix A for their full psychometric profiles.

This subsection reviews six studies evaluating adaptations of the Fear of Progression Questionnaire (FoP-Q) [33], with a predominant focus on the 12-item short form (FoP-Q-SF) originally developed by Mehnert et al. [32] to enhance clinical feasibility. Five studies examined translated versions of the FoP-Q-SF in Simplified Mandarin, Malay, Traditional Chinese, Portuguese, and German, all of which received Category A ratings under COSMIN criteria. One study evaluated a Persian-language version of the original 43-item FoP-Q, which was rated Category B due to insufficient structural validity. See Table 4 and Appendix A for detailed psychometric profiles.

Mahendran et al.—FoP-Q-SF (Mandarin and English) [34]: Mahendran et al. validated both the Mandarin and English versions of the FoP-Q-SF among Singaporean cancer survivors. The Mandarin version was adapted using forward–backward translation, expert review, and pilot testing, confirming content validity. Structural validity was supported through CFA with excellent fit indices (CFI = 0.97; RMSEA = 0.054; SRMR = 0.039). Internal consistency was high across both language versions (α = 0.70–0.94), and test–retest reliability exceeded COSMIN thresholds (r = 0.83–0.85). Construct validity was demonstrated through strong correlations with HADS-Anxiety (r = 0.72), HADS-Depression (r = 0.53), and EORTC physical functioning (r = –0.38). Criterion validity was further supported via correlations with the FCRI (r = 0.66) and FRQ (r = 0.64). However, measurement error, cross-cultural validity, and responsiveness were not assessed. Despite these omissions, the PROM demonstrated sufficient evidence for content, structural, internal consistency, and criterion validity, meeting COSMIN criteria for Category A. The FoP-Q-SF is recommended for use in research and clinical screening among Mandarin- and English-speaking cancer survivor populations, though additional studies could further confirm cross-cultural equivalence and responsiveness to change.

Abd Hamid et al.—FoP-Q-SF-M (Malay) [35]: Abd Hamid et al. translated and validated the FoP-Q-SF into Malay (FoP-Q-SF-M) among 200 cancer survivors in Malaysia. The translation followed Beaton’s cross-cultural adaptation procedure [55], including forward–backward translation, expert review, and pretesting with 20 native Malay speakers. Structural validity was rated sufficient based on exploratory factor analysis that met all COSMIN criteria: KMO = 0.900, significant Bartlett’s test (*p* < 0.001), one-factor solution with an eigenvalue of 6.69 explaining 55.75% of variance, and item loadings exceeding 0.65. Internal consistency was excellent (α = 0.93), and test–retest reliability was adequate (ICC = 0.89). Construct validity was supported through convergent correlations with the total score (r = 0.65–0.81) and low correlations with CTSQ-M domains for discriminant validity (r = –0.23 to 0.28). Criterion validity was not assessed, and no data were provided for measurement error, responsiveness, or cross-cultural validity. Based on COSMIN criteria, the FoP-Q-SF-M was rated Category A and is recommended for cross-sectional research and clinical screening use in Malay-speaking cancer populations.

Youssef et al.—FoP-Q-RS (German) [36]: Youssef et al. developed and validated the FoP-Q-RS, a five-item short form of the FoP-Q-SF [32], in a sample of 1002 long-term cancer survivors recruited through the Leipzig Cancer Registry in Germany. The instrument was designed to screen for core domains of fear of progression—affective response, family impact, occupational concerns, and loss of autonomy—while maintaining brevity for clinical use. Items were selected through expert consensus among four researchers, including one of the original FoP-Q-SF developers. Although this ensured coverage of key conceptual domains, no patient input or cognitive debriefing was reported. Structural validity was rated sufficient, based on CFA (CFI = 0.936; SRMR = 0.048). Internal consistency was supported by composite reliability (CR = 0.793), and criterion validity was demonstrated using ROC analysis against the GAD-7 (AUC = 0.79), with sensitivity of 72% and specificity of 70% at the optimal cut-off of ≥12. Convergent validity was supported through a moderate correlation with GAD-7 (r = 0.53, *p* < 0.001). No data were available on test–retest reliability, cross-cultural validity, measurement error, or responsiveness. The FoP-Q-RS meets COSMIN standards for content, structural, internal consistency, and criterion validity and qualifies for Category A. It is recommended for research and clinical screening use but is not yet suitable for longitudinal monitoring due to the absence of responsiveness data.

Cheng et al.—FoP-Q-SF Traditional Chinese [37]: Cheng et al. validated the Traditional Chinese version of the 12-item FoP-Q-SF in a sample of 311 Hong Kong Chinese cancer survivors who had completed treatment within the past 5 years. The aim was to assess its psychometric properties for use in Traditional Chinese-speaking populations. Structural validity was rated sufficient based on CFA (CFI = 0.954; SRMR = 0.042; RMSEA = 0.073), supporting a unidimensional model with two pairs of correlated item errors. Internal consistency was excellent (α = 0.922), with corrected item-total correlations ranging from 0.37 to 0.78. Construct validity was supported by both convergent correlations with five domains of the SCNS-SF34, established through Spearman’s correlations (r_S_ = 0.339–0.816), and known-group testing (e.g., higher FoP scores among women, younger participants, and those who had received chemotherapy or radiotherapy). However, content validity was rated indeterminate, as the scale was adopted from an earlier translation based on an unpublished master’s thesis and did not undergo cognitive interviewing or expert review. Criterion validity was also indeterminate due to the SCNS-SF34 not being conceptually aligned with fear of progression. No data were reported for test–retest reliability, cross-cultural validity, measurement error, or responsiveness. Based on COSMIN criteria, this version was rated Category A, supported by sufficient structural validity and internal consistency with at least low-quality evidence for content validity. It is recommended for cross-sectional use in Traditional Chinese-speaking populations but requires further validation for longitudinal monitoring or cross-cultural equivalence.

Silva et al.—FoP-Q-SF Portuguese [38]: Silva et al. translated and validated the FoP-Q-SF for use in Portuguese-speaking cancer survivors, with the study conducted in a sample of 220 adults in Portugal. The sample was primarily composed of women with breast cancer (74%), with a mean age of 48.8 years. Content validity was supported through forward–backward translation, expert review, and pretesting with survivors; however, the methodological detail was limited, leading to a low quality of evidence rating. Structural validity was supported via CFA of a modified one-factor model, which met COSMIN thresholds (CFI = 0.96; RMSEA = 0.08; WRMR = 0.80). Internal consistency was strong (Cronbach’s α = 0.86). Construct validity was demonstrated through moderate-to-strong correlations with psychological distress and quality of life domains, especially anxiety (r = 0.69) and emotional functioning (r = 0.58), although some hypothesized negative correlations with QoL were weaker than expected. Criterion validity was indeterminate, as no ROC analyses were performed and all correlations with comparators remained below the COSMIN ≥ 0.70 threshold. No data were reported for test–retest reliability, measurement error, cross-cultural validity, or responsiveness. Based on COSMIN criteria, this version was rated Category A, supported by sufficient content validity, structural validity, and internal consistency. It is recommended for cross-sectional use in clinical and research settings in Portuguese-speaking populations but requires further testing before being used to monitor change over time.

Hasannezhad Reskati et al.—FoP-Q Persian [39]: Hasannezhad Reskati et al. translated and validated the original 43-item FoP-Q for use with Persian-speaking cancer patients. The study involved 430 adults with gastrointestinal cancers recruited from two oncology centers in Northern Iran. The translation followed WHO forward–backward procedures, with expert review and qualitative feedback from 10 patients. Quantitative content validity was rated sufficient (CVI = 0.95; CVR = 0.78). Structural validity was rated insufficient: exploratory factor analysis yielded a five-factor structure (emotional response, employment, loss of independence, economy/family, and coping) explaining only 37.2% of the total variance, below COSMIN’s ≥ 50% threshold for sufficient structural validity. Internal consistency was acceptable across most subscales (α = 0.71–0.83), but due to insufficient structural validity, internal consistency could not be rated sufficient. Construct validity was supported through hypothesis testing using the HADS, with significant positive correlations observed for anxiety (r = 0.68) and depression (r = 0.55). These data were clearly reported in Table 3, though not discussed in the narrative text. Criterion and cross-cultural validity were not assessed. Test–retest reliability was evaluated, with ICCs reported above 0.70, though without detailed metrics per subscale. Responsiveness and measurement error were not evaluated. Based on COSMIN criteria, this PROM was rated Category B, reflecting a promising instrument with good initial translation and reliability, but lacking full validation evidence, particularly regarding structural and construct validity. It is not currently recommended for routine clinical or research use until further testing is completed.

### 3.3. New Developed FCR PROMs

This section presents PROMs created after 2012 that were not derived from existing instruments. These measures were designed to capture core FCR domains with fewer items, enhanced feasibility, or new conceptual approaches. Subsections cover (a) the Cancer Worry Scale (CWS) and its adaptations, (b) brief screeners such as the FCR4 and FCR7, (c) the Concerns About Recurrence Questionnaire (CARQ), and (d) the one-item FCR-1 and its adaptations. See Table 3 and Appendix A for their descriptive characteristics and full psychometric profiles.

(a)Cancer Worry Scale (CWS) [40]—Validation and Adaptation of the New PROM in New Languages and Populations

This subsection reviews studies that evaluated translated or adapted versions of the CWS, a brief measure of cancer-specific worry originally developed for hereditary cancer contexts and later validated in broader oncology populations. Three studies assessed its psychometric performance in Dutch-, English-, and Italian-speaking samples of cancer survivors. Each version is evaluated individually below. See Table 4 and Appendix A for their full psychometric profiles.

Custers et al.—CWS-8 Dutch [40]: Custers et al. validated the Dutch eight-item Cancer Worry Scale (CWS) in a sample of 194 female breast cancer survivors assessed up to 11 years post-treatment. Exploratory factor analysis confirmed unidimensionality (55.2% variance explained), and internal consistency was high (α = 0.87). Criterion validity was evaluated through ROC analysis using a two-item Cancer Acceptance Scale as a gold standard. The CWS demonstrated screening utility with a cut-off ≥ 12 (sensitivity = 96%, specificity = 56%) and a diagnostic threshold of ≥14 (sensitivity = 77%, specificity = 81%). Convergent and divergent validity were supported through expected correlations with fatigue and empowerment. The PROM met COSMIN criteria for sufficient structural validity, internal consistency, construct and criterion validity, and was rated Category A.

Custers et al.—CWS-6 English [41]: In a pooled sample of 981 cancer patients and survivors from five studies (breast, prostate, colorectal, and GIST cancers), Custers et al. re-validated the six-item English version of the CWS. CFA supported unidimensionality with good fit (CFI = 1.00, TLI = 1.00, RMSEA = 0.08). Internal consistency was excellent (α = 0.90). Construct validity was supported through moderate to strong correlations with the FCRI (r = 0.81), HADS-Anxiety (r = 0.64), and IES-Intrusion (r = 0.66). ROC analyses established cut-offs for high (≥10) and severe (≥12) FCR based on FCRI-SF thresholds, yielding AUCs between 0.90 and 0.93. These findings supported discriminative validity. The PROM was rated Category A, with strong evidence for its utility as a brief screener across cancer populations.

Chirico et al.—CWS-8 Italian [42]: Chirico et al. adapted and validated the Italian version of the eight-item CWS [40] in a sample of 226 breast cancer survivors. A two-factor solution (“cancer worries” and “worry impact”) was supported by CFA with adequate fit (TLI = 0.96, RMSEA = 0.07), and 63% of variance explained. Internal consistency was high (α = 0.90), with strong reliability for both subscales. Construct validity was supported via convergent correlations with HADS-Anxiety (r = 0.654) and the FCRI (r = 0.80). ROC analysis against the CAS yielded an AUC = 0.871 and optimal cut-off ≥16 (sensitivity = 74%, specificity = 85%). Despite no DIF or formal cross-cultural analysis, the overall psychometric profile justified a Category A rating for use in Italian breast cancer populations.

(b)FCR4 and FCR7 [43]—Validation and Adaptation of the New PROM in New Languages and Population

This subsection reviews studies evaluating the psychometric properties of the FCR4, FCR7, and their derivatives across diverse cancer populations. Originally developed in English to provide a brief yet valid screening option for fear of cancer recurrence, these tools have since been adapted and tested in Simplified Chinese, Traditional Chinese, Tamil, Spanish, German, and Brazilian Portuguese. Across studies, versions varied in sample size, clinical context, and psychometric scope. Each version is evaluated individually below. See Table 4 and Appendix A for their full psychometric profiles.

Humphris et al.—FCR4 and FCR7 (English) [43]: Humphris et al. developed the FCR4 and FCR7 as brief, unidimensional PROMs for assessing FCR in cancer survivors. The study included 259 English-speaking participants (206 breast and 53 colorectal cancer patients). Items were derived from established FCR scales and refined through expert consensus and piloting with patients. Exploratory factor analysis supported a one-factor solution for both tools, with KMO values of 0.86 (FCR4) and 0.92 (FCR7), and eigenvalues of 3.3 and 4.8, respectively, explaining over 50% of the variance. Factor loadings were all > 0.70 except one at 0.52, retained for conceptual relevance. Internal consistency was excellent (α = 0.93 for FCR4; α = 0.92 for FCR7). Construct validity was confirmed through significant correlations with anxiety and depression (FCR4: r = 0.65 and r = 0.36; FCR7: r = 0.68 and r = 0.40), and criterion validity was supported via ROC analysis using HADS-Anxiety (Cohen’s d = 1.87 for FCR4). However, no data were reported on test–retest reliability, measurement error, responsiveness, or cross-cultural validity. Based on COSMIN criteria, both versions meet Category A requirements due to sufficient content validity, structural validity, and internal consistency. These tools are recommended for use in English-speaking cancer populations, with either version suitable depending on clinical context. The authors noted that, from a psychometric perspective, there is “little to choose between the FCR4 and FCR7,” and decisions regarding use may be guided by context (e.g., screening vs. monitoring over time).

Yang et al.—FCR7 Simplified Chinese [44]: Yang et al. translated and validated the FCR7 into Simplified Chinese in a large sample of 1025 cancer patients in China. Content validity was supported through forward–backward translation and expert panel review, with a reported CVI of 0.88. Structural validity was confirmed using both exploratory factor analysis (65.4% variance explained) and CFA (CFI = 0.996; RMSEA = 0.039; χ^2^/df = 1.79). Internal consistency was strong (α = 0.87), and test–retest reliability was rated sufficient (r = 0.90). Construct validity was supported by significant correlations with the FoP-Q-SF (r = 0.756), PHQ-9 (r = 0.522), and GAD-7 (r = 0.553). Criterion validity was indirectly confirmed through these same associations. No data were available for measurement error, cross-cultural validity, or responsiveness. The scale was rated COSMIN Category A and is recommended for use among Chinese-speaking oncology populations in cross-sectional settings.

Lee et al.—FCR7-C [45]: Lee et al. evaluated the psychometric properties of the FCR7-C in a sample of 160 early-stage lung cancer patients in Taiwan. Content validity was confirmed through expert review and patient feedback. Structural validity was supported by CFA (CFI = 0.97; SRMR = 0.04), affirming a unidimensional structure. Internal consistency was strong (α = 0.90), and item-total correlations were acceptable. Construct validity was demonstrated via expected correlations with anxiety, depression, and quality of life measures, and criterion validity was confirmed through correlations with emotional distress indicators. Test–retest reliability, measurement error, cross-cultural validity, and responsiveness were not evaluated. The FCR7-C received a COSMIN Category A rating and is recommended for screening purposes in Chinese-speaking lung cancer populations.

Braun et al.—FCR6-Brain (German) [46]: Braun et al. adapted the FCR7 into a six-item version, the FCR6-Brain, for use in patients with primary brain tumors. The validation was conducted in a German-speaking sample of 86 adult patients. Content validity was ensured through systematic item development guided by expert and patient input. Structural validity was rated sufficient, with a one-factor solution explaining 80.5% of variance and item loadings all above 0.80. Internal consistency was excellent (α = 0.91). Construct validity was confirmed through correlations with GAD-7 (r = 0.70), PHQ-9 (r = 0.50), and the DDS (r = 0.70). Criterion validity was indirectly supported through these correlations, as ROC analyses were not performed. The authors proposed clinically meaningful cut-offs based on percentile ranks, reflecting the higher burden of FCR in neuro-oncology. No data were available for test–retest reliability, cross-cultural validity, or responsiveness. Based on COSMIN criteria, the FCR6-Brain was rated Category A and is recommended for clinical use in brain tumor populations.

Iglesias-Puzas et al.—FCR7 Spanish [47]: Iglesias-Puzas et al. translated and validated the FCR7 in a sample of 119 Spanish-speaking melanoma patients. The cross-cultural adaptation followed Beaton’s guidelines [55], including forward–backward translation, expert review, and cognitive debriefing with 15 patients. Structural validity was not confirmed, as no CFA or model fit indices were reported. Internal consistency was acceptable (α = 0.834) but could not be rated sufficient due to missing structural validity evidence. Construct validity was partially supported through one predefined hypothesis confirmed via correlation with SCI-12 emotion and appearance subscales. Criterion validity was indeterminate, as the SCI-12 is not closely aligned with FCR and no ROC analysis was conducted. Test–retest reliability, measurement error, and responsiveness were not assessed. The FCR7-Spanish was rated COSMIN Category B and is considered promising but not yet recommended for widespread clinical use without further validation.

Nandakumar et al.—FCR7 Tamil [48]: Nandakumar et al. translated and evaluated the Tamil version of the FCR7 [43] among 106 breast cancer survivors. The translation process followed international standards including dual forward–backward translation, expert review, and face validity testing with 10 patients. Content validity was rated sufficient. Structural validity was not evaluated (no factor analysis conducted), and thus internal consistency (α = 0.864) could not be formally rated according to COSMIN. Test–retest reliability was strong, with ICC = 0.910 over a 15-day interval in a subset of 32 participants. Construct validity was supported through theoretically aligned correlations with the FACT-B (r = −0.259, *p* = 0.01) and IES-R (r = 0.270, *p* = 0.01), and criterion validity was rated sufficient based on these same measures. Measurement error, cross-cultural validity, and responsiveness were not assessed. Based on COSMIN guidelines, the FCR7-Tamil meets criteria for Category B due to the absence of structural validity and resulting indeterminate internal consistency. This version shows promise for use in Tamil-speaking breast cancer populations but requires further validation before it can be recommended for routine clinical application.

Bergerot et al.—FCR4 and FCR7 Brazilian Portuguese [20]: Bergerot et al. translated and validated the FCR4 and FCR7 in a Brazilian Portuguese-speaking sample of 200 adult cancer survivors. The adaptation process followed Beaton’s cross-cultural methodology [55], including forward–backward translation, expert panel review, and cognitive debriefing. CFA supported unidimensionality for both tools, with excellent fit for the FCR4 (CFI = 1.00; RMSEA = 0.0001; SRMR = 0.0001) and acceptable fit for the FCR7 (CFI = 0.971; RMSEA = 0.099; SRMR = 0.036). Internal consistency was high (α = 0.879 for FCR4; α = 0.894 for FCR7). However, test–retest reliability, construct and criterion validity, cross-cultural validity, and responsiveness were not reported. Due to sufficient content validity, structural validity, and internal consistency, the PROMs were rated Category A and are recommended for screening use in Brazilian Portuguese-speaking cancer populations, with further research needed to confirm responsiveness and construct validity.

(c)Concerns About Recurrence Questionnaire (CARQ) CARQ-4 [49]—Validation of the New PROM in a New Language and Populations

This subsection reviews the development and validation of the CARQ-4, a newly created, brief measure designed to screen for fear of cancer recurrence among breast cancer survivors. See Table 4 and Appendix A for its full psychometric profile.

Thewes et al.—CARQ-4 [49]: Thewes et al. developed and validated the Concerns About Recurrence Questionnaire–4 (CARQ-4), a brief, unidimensional screening tool for fear of cancer recurrence among breast cancer survivors. The tool was tested in two samples: 218 Australian women diagnosed before age 45 and 2001 Danish women aged 26–70. Content validity was supported through expert review and pre-testing with survivors, although no cognitive interviewing was reported. Structural validity was confirmed through exploratory factor analysis in the Australian sample, where the CARQ-4 explained 72% of the variance, and CFA in the Danish sample (CFI = 0.99; RMSEA = 0.12). Internal consistency was strong in both samples (α = 0.87 Australia; α = 0.88 Denmark), and test–retest reliability exceeded COSMIN thresholds (r = 0.74 and 0.83, respectively). Construct validity was confirmed through correlations with the FCRI-Total and Severity subscales (r = 0.76–0.78) and moderate associations with anxiety, depression, and health anxiety. Criterion validity was supported by ROC analysis against the FCRI-Severity, yielding an AUC = 0.90 and optimal cut-off score of ≥ 12 (sensitivity = 85%, specificity = 81%). However, Rasch analysis revealed DIF by age and language, indicating insufficient cross-cultural validity for the Danish version. Measurement error and responsiveness were not evaluated. Based on COSMIN criteria, the English CARQ-4 received a Category A rating and is recommended as a brief screening tool for FCR among breast cancer survivors. The Danish version does not meet current psychometric standards and is not recommended for use.

(d)FCR-1 [50]—Validation of a New PROM, FCR-1, and Adaptations in New Languages and Populations

This subsection presents three single-item PROMs developed or adapted to screen for FCR: the original FCR-1 by Rudy et al. [50], the FCR-1r by Smith et al. [51], and a Danish-Language version of the FCR-1r validated in a population-based cohort by Lyhne et al. [52]. All tools are brief, modeled on the Edmonton Symptom Assessment System (ESAS) structure, and designed for use in time-limited clinical contexts. See Table 4 and Appendix A for their full psychometric profiles.

Rudy et al.—FCR-1 [50]: Rudy et al. developed and validated the FCR-1, a single-item PROM developed to efficiently screen for FCR using a 0–100 visual analogue scale (VAS). The validation was conducted in an English-speaking sample of 69 breast and gynecological cancer survivors. Participants were asked “On a scale from 0 to 100, what is your subjective level of fear of cancer recurrence at this time?” Although the published version specified a 0–100 format, respondents were also permitted to use a 0–10 scale if preferred—a detail clarified through personal communication with the lead author of the FORT randomized controlled trial [56], which provided the validation data. Rudy et al. have that their one-item FCR measure was modeled after the Edmonton Symptom Assessment System (ESAS) for cancer symptoms [50]. Thus, the reason to offer respondents to complete either as a percentage or 0–10 scale. Content validity was supported by expert review and patient feedback. Construct validity was sufficient, with all a priori hypotheses confirmed and expected correlations observed with the FCRI (r = 0.395), illness uncertainty (r = 0.493), and reassurance-seeking (r = 0.325). Criterion validity was demonstrated using the FCRI-Severity (cut-off ≥ 22) as reference (AUC = 0.85; sensitivity = 70%; specificity = 89.5%). Responsiveness was confirmed through repeated-measures analyses across six sessions, with a large effect size (d = 0.79) from the weekly sessions 1 to 6. Structural validity and internal consistency were not applicable as this is a single-item PROM. Test–retest reliability and cross-cultural validity were not assessed. The FCR-1 [50] met COSMIN criteria for Category A and is recommended for use as both a screening and longitudinal monitoring tool. Its key strengths include sufficient validity across multiple domains and high responsiveness. Limitations include the absence of test–retest and cross-cultural validation. See Table 4 and Appendix A for full details.

Smith et al.—FCR-1r [51]: Smith et al. [51] developed a single-item PROM, the FCR-1r, adapted from the ESAS format to assess “fear of recurrence or progression” on a 0–10 numeric rating scale. Validation was conducted in an Australian sample of 107 adult cancer survivors. The tool was refined through consultation with survivors and clinicians, including a clarifying phrase to enhance comprehension. Content validity was rated sufficient, although the quality of evidence was downgraded due to limited detail on cognitive testing procedures and the absence of consultation with the original FCR-1 developers [50]. Such as, while the FCR-1r was introduced as a “revision” of the original FCR-1 [50], the original FCR-1 developers were not involved in the adaptation process, and the revised item, which features different wording and a numeric scale format, was not tested in parallel with the original version. While adaptation and refinement are important steps in tool development, international guidelines (e.g., COSMIN, ISPOR, Beaton, WHO) encourage collaborative processes with original scale developers to preserve conceptual continuity and ensure construct validity in modified PROMs. As for construct validity of the FCR-1r, it was supported through hypothesis testing with strong correlations to the FoP-Q-SF (r = 0.67), anxiety (r = 0.54), depression (r = 0.52), and distress (r = 0.57). Criterion validity was demonstrated using the FCRI-SF as a comparator (cut-off ≥ 22), yielding an AUC = 0.91 and a suggested cut-off of ≥ 5/10 (sensitivity and NPV > 90%, specificity ~80%). Because one of the datasets was collected during routine care, the methodological quality for hypothesis testing and criterion validity was rated as adequate, resulting in a moderate overall evidence grade. Structural validity and internal consistency were not applicable, and test–retest reliability, cross-cultural validity, and responsiveness were not assessed. The FCR-1r met COSMIN criteria for Category A and is recommended for cross-sectional screening. However, it is not yet validated for longitudinal monitoring, and additional research is needed to assess its stability and responsiveness over time. See Table 4 and Appendix A for full details.

Lyhne et al.—FCR-1r Danish [52]: Lyhne et al. developed through translation and field-testing a Danish version of the FCR-1r in a large population-based sample of long-term colorectal cancer survivors (*n* = 1654), with involvement from the original FCR-1r by Smith et al. [51] authors. Content validity was rated sufficient based on a rigorous trilingual translation process and cognitive debriefing with survivors. Criterion validity was established using the FCRI-SF (cut-off ≥ 22), with the FCR-1r achieving an AUC = 0.93 (95% CI: 0.91–0.94), and a ≥ 5/10 cut-off yielding 93.5% sensitivity and 80.4% specificity. Construct validity was supported by a priori hypotheses, with expected associations to anxiety, depression, distress, and quality of life. Structural validity and internal consistency were not applicable as this is a single-item tool. Measurement error, responsiveness, and test–retest reliability were not assessed.

The FCR-1r Danish met COSMIN criteria for Category A and is recommended for cross-sectional screening in long-term colorectal cancer survivors. However, it is not yet validated for longitudinal use, and further research is needed to assess responsiveness and temporal stability. See Table 4 and Appendix A for full details.

## 4. Discussion

This systematic review evaluated 34 distinct PROMs across 32 studies using the COSMIN framework to assess the quality of instruments used to measure FCR. Through a rigorous appraisal of content and structural validity, reliability, and other psychometric properties, this review offers a comprehensive synthesis of the current landscape of FCR measurement. Building on the foundational review by Thewes et al. [6], which identified 20 multi-item FCR measures and highlighted the need for further psychometric refinement, the present review captures PROMs developed or validated between 2011 and 2023 and reflects a more mature, diverse, and internationally responsive measurement field.

While Thewes et al. relied on general psychometric criteria, this review applies the full COSMIN methodology—including risk of bias assessment, criteria for good measurement properties, and GRADE-modified evidence ratings. PROMs were systematically categorized as A, B, or C, providing actionable guidance for research and clinical practice. The review also evaluates cross-cultural validity and addresses the proliferation of brief and ultra-brief instruments, including single-item measures like the FCR-1. These methodological enhancements ensure that the current review represents not just an update, but a rigorous and clinically relevant advancement aligned with contemporary psychometric standards.

### 4.1. Summary of Main Findings

Of the 34 PROMs evaluated across 32 report studies, 28 instruments met COSMIN criteria for Category A, indicating they are supported by sufficient psychometric evidence for use in clinical or research settings. These include widely established measures such as the full FCRI [5], validated in multiple language and population contexts, including English [21], Mandarin [25], Korean [22], Dutch [23], and Chinese long version [19], as well as the CARS, validated in Japanese (CARS-J) [26].

Shortened forms and screening variants of the FCRI also performed well, with Category A ratings assigned to the French FCRI-SF [29], English version [30], Brazilian Portuguese version [20], Chinese shorter version [19], and Turkish shorter form [28].

From the FoP-Q family, five short forms were successfully validated in new languages and populations: Simplified Mandarin [34], Malay [35], Traditional Chinese [37], Portuguese [38], and German [36]. All five achieved COSMIN Category A ratings and are recommended for cross-sectional screening.

Among newly developed instruments, twelve PROMs met Category A standards, including the CWS-6 and CWS-8 [40,41], the CARQ-4 [49], the FCR4 and FCR7 [43] and their validated adaptations in Chinese [44,45] and Portuguese [20], the FCR6-Brain [46], and the FCR-1 [50].

Finally, two single-item adaptations of the FCR-1, the FCR-1r [51] and the Danish FCR-1r [52], were also classified as Category A for cross-sectional screening. However, neither assessed responsiveness or test–retest reliability, only the FCR-1 [50] did assess, and is the only one that can be recommended for longitudinal monitoring at this time.

Together, these 28 PROMs demonstrated sufficient structural validity, internal consistency, and construct validity. Several, including the FCRI English version [21], K-FCRI [22], FCR-1 [50], and the various FoP-Q-SF translations, offer growing international applicability and are well-positioned for integration into clinical and research pathways addressing fear of cancer recurrence.

Together, the 34 PROMs evaluated in this systematic review represent a significant expansion of the measurement landscape for FCR. By applying the full COSMIN framework, this review offers a comprehensive appraisal of the psychometric robustness of both established and newly developed tools. The following sections synthesize key findings by PROM type and validation context, highlighting methodological strengths, persistent gaps, and implications for the clinical implementation of FCR assessment tools.

Established PROMs with Cultural or Population-Specific Validation: Multiple high-quality cultural validations of the original FCRI [5] were identified, including English [21], Mandarin [25], Korean [22], Dutch [23], and Chinese [19]. These studies consistently demonstrated sufficient structural validity, internal consistency, and construct validity, qualifying them for COSMIN Category A. The Japanese version of the CARS [26] also showed sufficient psychometric performance despite lacking formal cross-cultural equivalence testing, and was rated Category A.

FCRI-Derived Short Forms and Screening Variants: Since the publication of the Thewes et al. [6] systematic review, one of the most notable advances in the FCR measurement field has been the development and validation of shorter-form PROMs designed for efficient clinical screening. The present review identifies validated screening variants stemming from established instruments (FCRI and FoP-Q) and a wave of newly developed ultra-brief tools, including multiple 1–7 item scales. Several short forms derived from the FCRI were evaluated. The 24-item Turkish adaptation [28] and the 10-item Chinese short form [19] both achieved Category A ratings. For screening versions, among the FCRI-derived short forms, the nine-item FCRI-SF has emerged as the most consistently validated screening tool across populations and languages. The French version [29] demonstrated strong criterion validity against a structured FCR interview (AUC = 0.88), with a recommended cut-off of ≥13 to identify clinically significant FCR. The English version tested for its screening capacity in Fardell [30] was validated against semi-structured interviews and yielded country-specific cut-offs (≥22 in the Canadian sample, ≥13 in Australia). The Brazilian Portuguese version [20] met COSMIN Category A criteria and is recommended for cross-sectional screening, though an optimal cut-off was not specified. In contrast, the Chinese version [31] was downgraded to Category C due to lack of structural validity and low specificity (35%), and should not be used until further validated.

FoP-Q and Its Adaptations: The FoP-Q-SF was successfully validated in five language and population groups, Simplified Chinese [34], Malay [35], Traditional Chinese [37], Portuguese [38], and German [36]. All received COSMIN Category A ratings and had a consistent screening threshold of ≥ 34 was supported across several studies. The newly developed FoP-Q-RS [36], a five-item rapid screener, also demonstrated promising performance (AUC = 0.79, cut-off ≥ 12) and was rated Category A, though test–retest reliability and responsiveness remain untested. All PROMs in Table 4, Category 2.3 (FoP-Q—Adaptations and Validations in New Languages and Populations) were rated Category A, except for the Persian full version [39], which had insufficient structural validity and was rated Category B.

Newly Developed PROMs and Their Translations: Several entirely new PROMs developed since 2014 now provide clinicians with diverse options for brief FCR screening. These include the CWS-6 and CWS-8 [40,41], the CARQ-4 [49], the FCR4 and FCR7 [43], and the brain cancer-specific FCR6-Brain [46]. All were classified as COSMIN Category A, supported by sufficient structural validity, internal consistency, and criterion validity. Specifically, The CWS [40,41,42] demonstrated consistent unidimensionality, excellent reliability, and discriminative ability across Dutch and Italian populations, qualifying all three versions as Category A. The FCR4/FCR7 tools [43] and their translated variants [20,44,45,46] also achieved Category A ratings, with strong evidence for structural and criterion validity. Their brevity and consistent psychometric strength support their use in diverse oncology contexts for both screening and clinical monitoring. Full property-level ratings are available in Table 4 and Appendix A. The Spanish [47] and Tamil [48] FCR7 adaptations were rated Category B due to limited evidence on structural validity. The CARQ-4 [49] demonstrated excellent screening accuracy in English, with strong evidence for structural and criterion validity. However, Rasch analysis revealed language-related DIF in the Danish version, limiting cross-national comparability.

Many also provided pragmatic or percentile-based cut-offs, such as ≥12 for the CWS-6 [41], and same for the CARQ-4 [49], and ≥17/27 (moderate/severe) for the FCR7, both for the English and Portuguese version [20,43]. While several were validated in breast and mixed cancer populations, others, such as the FCR6-Brain [46] and CWS-6 [41], extend the evidence base to include patients with primary brain tumors or gastrointestinal cancers.

### 4.2. Ultra-Brief and Single-Item Tools

Among these new instruments, the FCR-1 [50] warrants special emphasis. As the only validated single-item PROM with evidence of responsiveness, the FCR-1 uniquely supports both screening and longitudinal monitoring. Validated in a sample of 69 women with breast and gynecological cancers, the tool demonstrated large responsiveness (Cohen’s d = 0.79) across six sessions of a cognitive-existential FORT intervention [56], and criterion validity against the FCRI-Severity (AUC = 0.85). COSMIN Category A was assigned with high-quality ratings for content validity, hypothesis testing, criterion validity, and responsiveness. A cut-off of ≥ 4.5 (on the 0–10 scale) is recommended for identifying high FCR. In contrast, two adaptations of the FCR-1, the FCR-1r [51] and the Danish FCR-1r [52], were also assigned Category A for cross-sectional screening but have not assessed responsiveness or test–retest reliability. As such, only the original FCR-1 should be used for repeated assessments over time.

The recent inclusion of the FCR-1r in the Ontario Health (CCO) clinical guideline as the preferred screening item for routine care [7] reflects growing enthusiasm for practical, ultra-brief measures in survivorship contexts. However, this recommendation appears to have preceded the completion of our systematic review. Our findings suggest that while both the FCR-1r [51,52] and the original FCR-1 [50] meet COSMIN Category A criteria for cross-sectional screening, only the original FCR-1 has been evaluated for responsiveness, showing robust sensitivity to change (effect size d = 0.79). As such, at present, the FCR-1 remains the only validated single-item tool suitable for both screening and longitudinal monitoring of FCR. The FCR-1r, while promising, would benefit from further psychometric evaluation, particularly responsiveness and test–retest reliability, to support its broader use in ongoing assessment. We hope these findings may inform future updates to guideline recommendations as the evidence base continues to evolve.

### 4.3. Strengths and Weaknesses of the Current Evidence

This review highlights notable strengths in the current body of FCR PROM validation literature. Many PROMs demonstrated evidence for sufficient content validity (any level), and sufficient internal consistently (at least low quality, meaning also sufficient structural validity), which COSMIN identifies as absolutely required for PROMs to be categorized as ‘A’, which means can be recommended for use and results obtained with these PROMs can be trusted [15,16,17]. Structural validity was confirmed in several studies via confirmatory or exploratory factor analysis with acceptable model fit, particularly for newer tools such as the CWS [41], FCR4/7 [43], and CARQ-4 [49]. In a number of screening tools, criterion validity was established through ROC analyses using structured interviews or reference measures.

However, important weaknesses persist. Structural validity was frequently untested or assumed in studies using translated or shortened PROMs. Several studies cited reliability or dimensionality data from prior validations in different populations or languages, limiting the ability to confirm psychometric performance within the study context. Content validity was often inadequately documented, with few studies reporting on cognitive interviewing or patient involvement. Responsiveness, measurement error, and cross-cultural validity remain among the most underreported properties across nearly all PROMs evaluated.

### 4.4. Implications for Clinical Practice and Research

The findings of this review offer a clear roadmap for clinicians and researchers selecting FCR PROMs. PROMs receiving Category A status, such as the FCRI-SF (English [30], French [29], Portuguese [20]), FoP-Q-SF (Simplified [34] and Traditional Chinese [37], Malay [35], Portuguese [20]), CWS (Dutch [40], English [41], Italian [42]), and the FCR4/7 [43], can be recommended for use in clinical screening and cross-sectional research. Ultra-brief tools such as the FCR-1 [50] and FCR-1r [51,52] may be especially valuable in settings where respondent burden must be minimized. However, among these, only the FCR-1 [50] demonstrated evidence of responsiveness, supporting its potential use for repeated assessment over time.

In practical terms, Category A PROMs can be regarded as the most robust options for clinical use, whereas Category B tools may be considered in settings where Category A instruments are unavailable, provided results are interpreted cautiously. Category C instruments should be avoided in routine clinical use until further psychometric validation is available. For clinicians seeking a brief screening option, the FCRI-SF and FoP-Q-SF represent strong choices validated across multiple languages and contexts, while the full FCRI and its adapted versions remain the most comprehensive assessments available. Appendix A and Table 4 provide structured comparisons to guide instrument selection based on clinical and research needs. For highlights the five PROMs with the strongest overall support, outlining their features, strengths, and limitations to aid clinicians in choosing appropriate tools for screening or longitudinal follow-up.

However, some caution is warranted in longitudinal applications and in cross-cultural contexts. Only a minority of PROMs provided sufficient evidence of responsiveness, and test–retest reliability was rarely evaluated. Furthermore, evidence of cross-cultural validity was often insufficient or indeterminate, largely due to the absence of differential item functioning (DIF) or multi-group confirmatory factor analysis (MG-CFA). This gap limits confidence in the comparability of scores across linguistic and cultural groups and constrains the global generalizability of many PROMs. In low-resource or language-restricted settings, Category B PROMs may be cautiously considered when Category A tools are unavailable, provided results are interpreted conservatively. Category C PROMs should generally be avoided in clinical practice until further validation is available, though they may hold exploratory value in contexts where no alternative instruments exist. To balance these limitations, Appendix A offers a concise overview of the five PROMs currently best supported by evidence, serving as a practical reference point for clinicians navigating tool selection. However, only a minority of PROMs provided sufficient evidence of responsiveness, and test–retest reliability was rarely evaluated, limiting their utility for monitoring FCR trajectories over time or evaluating intervention effects. Future research should therefore prioritize longitudinal designs, rigorous cross-cultural measurement equivalence testing, and PROM integration into clinical workflows to support equitable and scalable implementation.

### 4.5. Strengths and Limitations of the Review

This review advances substantially on prior review summaries [6], including a recent review by Díaz-Periáñez et al. [57]. Although that review references COSMIN guidance, it does not apply the methodology in full. PROMs were not evaluated individually by measurement property, and COSMIN’s evidence grading and A/B/C categorization were not implemented, limiting interpretability and practical utility. Additionally, the scope of their review was narrowly focused on the cross-cultural adaptation of only three FCR PROMs (FCRI-SF, FoP-Q-SF, and CARS), rather than offering a comprehensive psychometric synthesis. In contrast, the present review employed a broader search strategy and applied the full COSMIN stepwise approach—including risk of bias assessment, criteria for good measurement properties, and GRADE-based evaluation of evidence quality. Each PROM was assessed independently from related versions to ensure content-specific judgment based solely on data explicitly reported within each source publication. This rigorous and reproducible process supports clear, evidence-based recommendations for instrument selection across diverse clinical and research settings.

Nonetheless, this review also has limitations. First, only studies published in English or with accessible English translations were included, which may have excluded high-quality validations published in other languages. Second, no formal meta-analysis was conducted, given the substantial heterogeneity across study designs, cancer populations, and psychometric analyses. Lastly, the strength of the COSMIN ratings is inherently constrained by the methodological quality and completeness of reporting in the primary studies, which often featured small sample sizes or lacked critical design elements (e.g., longitudinal follow-up, cross-cultural DIF testing).

Although this review was finalized in 2025, the literature search was limited to publications through 31 December 2023. This cut-off aligns with COSMIN guidance on feasibility in PROM reviews [58], which emphasizes the importance of workload manageability when evaluating multiple measurement properties across numerous PROMs. The review ultimately included 34 distinct PROMs across 32 studies, exceeding COSMIN’s feasibility threshold of 25 PROMs. This decision ensured that the methodological rigor of the review could be maintained without compromising depth or accuracy.

### 4.6. Thematic Gaps in FCR PROM Research

Despite progress in FCR PROM development, several thematic gaps remain. First, the field lacks consensus on minimum clinically important differences (MCIDs) for many PROMs, limiting interpretability of change scores. Second, cultural adaptations often fall short of COSMIN standards for cross-cultural validity, with few studies conducting item-level DIF or invariance testing. Third, FCR PROMs are rarely co-developed with survivors, and few studies explicitly draw on patient narratives or lived experience frameworks.

Additionally, there is limited coverage of FCR in populations beyond early-stage survivors, such as patients with advanced disease, underrepresented ethnocultural groups, or sexual and gender minorities. Tools with adequate performance in one group may not generalize across care settings or cultures. Addressing these gaps will require coordinated methodological efforts, survivor involvement, and investment in cross-context validation research.

## 5. Conclusions

This systematic review provides the most comprehensive and methodologically rigorous evaluation to date of PROMs assessing fear of cancer recurrence. Applying COSMIN standards across 35 PROMs, we identified 28 PROMs that meet Category A criteria and can be recommended for screening or cross-sectional use, with a smaller subset supported for longitudinal monitoring. Brief tools such as the FCRI-SF, FCR-1, CWS-6, and CARQ-4 offer efficient screening options, while full-length scales like the FCRI and FoP-Q remain valuable for more comprehensive assessment. However, critical gaps persist in structural validity reassessment for translated instruments, in the evaluation of responsiveness, and in the representation of diverse populations. Future FCR PROM development and validation should prioritize cross-cultural testing, responsiveness studies, and conceptual frameworks that reflect lived experience. Clinicians and researchers now have a clear evidence base to guide PROM selection, and future clinical guidelines should incorporate these findings to optimize equitable, person-centered survivorship care.

## Figures and Tables

**Figure 1 healthcare-13-02165-f001:**
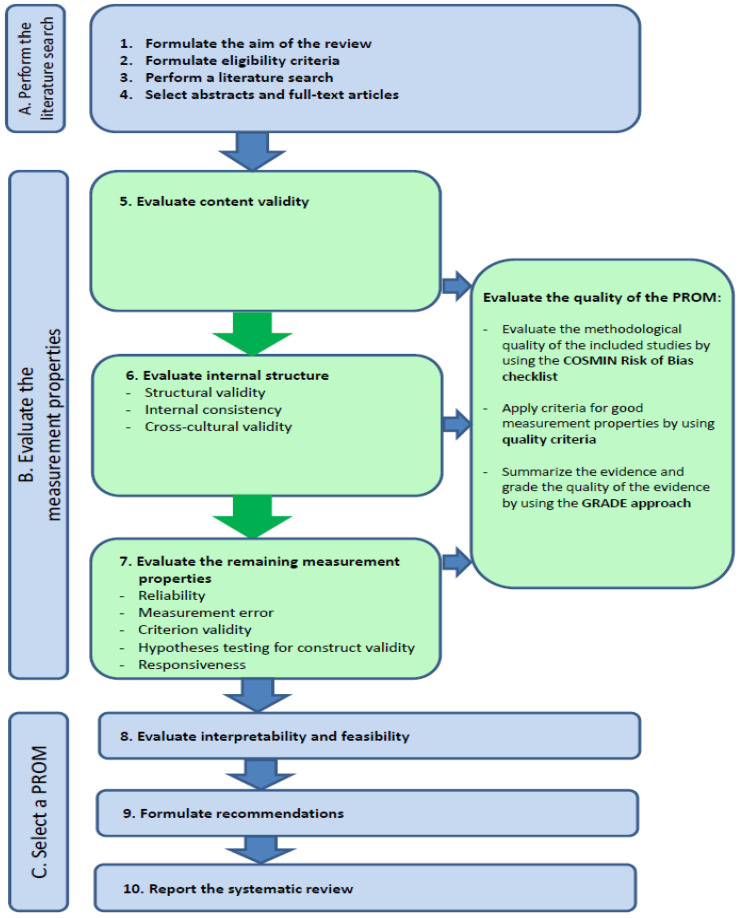
Ten steps for conducting a systematic review of PROMs [15,16].

**Figure 2 healthcare-13-02165-f002:**
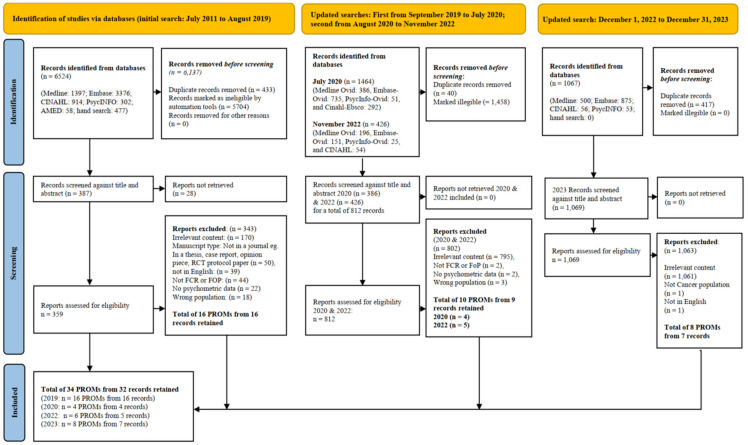
PRISMA 2020 flow diagram summarizing the selection process of included studies (July 2011–December 2023) [18].

**Table 1 healthcare-13-02165-t001:** Search terms.

Search Terms		
Cancer	Fear of Recurrence	Measure
(cancer* or neoplasm* or tumor?* or malignan* or carcinoma* or adenocarcinoma* or choriocarcinoma* or leuk?emia* or sarcoma* or teratoma* or hodgkin* or nonhodgkin* or metasta* or lymphoma* or melanoma* or myeloma* or oncolog* or glioblastoma* or multiforme or glioma* or oligodendroglioma* or astrocytoma* or meningioma*).	((fear or fears or anxiety* or anxious* or worry or worries or concern* or stress* or distress*) adj11 (recur?* or relaps* or progression)).	(questionnaire* or survey* or psychometric* or self report* or validat* or reliability or validity or sensitivity or specificity or reproducibility or reproducible or measur* or rate or rating or scale* or inventory or inventories or model*)

**Table 2 healthcare-13-02165-t002:** COSMIN criteria for good measurement properties [15].

Measurement Property	Rating ^1^	Criteria
Structural validity (mp1)	+	**CTT:** CFA: CFI or TLI **OR** comparable measure > 0.95 **OR** RMSEA < 0.06 **OR** SRMR < 0.08 ^2^ EFA: Kaiser Criterion KMO ≥ 0.80 **OR** Bartlett’s Test of Sphericity: *p* < 0.05 **OR** Eigenvalue ≥ 1 (for each retained factor) **AND** Variance explained ≥ 50% (preferably ≥ 60%) **AND** Factor loadings ≥ 0.40 **AND** Cross-loadings ≤ 0.20 (difference between highest and other loadings) **IRT/Rasch:** **No** violation of unidimensionality ^3^: CFI or TLI or comparable measure > 0.95 **OR** RMSEA < 0.06 OR SRMR < 0.08 **AND** **No** violation of local independence: residual correlations among the items after controlling for the dominant factor < 0.20 OR Q3’s < 0.37 **AND** **No** violation of monotonicity: adequate looking graphs OR item scalability > 0.30 **AND** Adequate model fit: **IRT: χ^2^ > 0.01** Rasch: infit and outfit mean squares ≥ 0.5 **AND** ≤ 1.5 **OR** Z standardized values > −2 **AND** < 2
	?	CTT: Not all information for ‘+’ reported IRT/Rasch: Model fit not reported
	−	Criteria for ‘+’ not met
Internal consistency (mp2)	+	At least low evidence ^4^ for sufficient structural validity ^5^ **AND** Cronbach’s alpha(s) ≥ 0.70 for each unidimensional scale or subscale
	?	Criteria for “At least low evidence for sufficient structural validity ^5^” not met
	−	At least low evidence ^4^ for sufficient structural validity ^5^ **AND** Cronbach’s alpha(s) < 0.70 for each unidimensional scale or subscale ^6^
Reliability (mp3)	+	ICC or weighted Kappa ≥ 0.70
	?	ICC or weighted Kappa not reported
	−	ICC or weighted Kappa < 0.70
Measurement error (mp4)	+	SDC or LoA < MIC ^5^
	?	MIC not defined
	−	SDC or LoA > MIC ^5^
Hypotheses testing for construct validity (mp5)	+	The result is in accordance with the hypothesis ^7^
	?	No hypothesis defined (by the review team)
	−	The result is not in accordance with the hypothesis ^7^
Cross-cultural validity/measurement invariance (mp6)	+	No important differences found between group factors (such as age, gender, language) in multiple group factor analysis **OR** no important DIF for group factors (McFadden’s R^2^ < 0.02)
	?	No MGCFA **OR** DIF analysis performed
	−	Important differences between group factors **OR** DIF were found
Criterion validity (mp7)	+	Correlation with gold standard ≥ 0.70 **OR** AUC ≥ 0.70
	?	Not all information for ‘+’ reported
	−	Correlation with gold standard < 0.70 **OR** AUC < 0.70
Responsiveness (mp8)	+	The result is in accordance with the hypothesis: before and after intervention ^7^ **OR** AUC ≥ 0.70
	?	No hypothesis defined (by the review team)
	−	The result is not in accordance with the hypothesis ^7^ **OR** AUC < 0.70

Notes: The criteria are from the COSMIN manual version 1.0 February 2018 and were said to be based on Prinsen et al. [15] and Terwee et al. [16]. AUC = area under the curve, CFA = confirmatory factor analysis, CFI = comparative fit index, CTT = classical test theory, DIF = differential item functioning, EFA = exploratory factor analysis, ICC = intraclass correlation coefficient, IRT = item response theory, LoA = limits of agreement, MIC = minimal important change; RMSEA = Root Mean Square Error of Approximation, SDC = smallest detectable change, SRMR = Standardized Root Mean Residuals, TLI = Tucker–Lewis index; χ^2^ chi-square likelihood ratio statistic (χ^2^). ^1^ “+” = sufficient, “−” = insufficient, “?” = indeterminate. ^2^ To rate the quality of the summary score, the factor structures should be equal across studies. ^3^ Unidimensionality refers to a factor analysis per subscale, while structural validity refers to a factor analysis of a (multidimensional) patient-reported outcome measure. ^4^ As defined by grading the evidence according to the GRADE approach. ^5^ This evidence may come from different studies. ^6^ The criterion ‘Cronbach alpha < 0.95’ was deleted, as this is relevant in the development phase of a PROM and not when evaluating an existing PROM. ^7^ The results of all studies should be taken together, and it should then be decided if 75% of the results are in accordance with the hypotheses.

**Table 3 healthcare-13-02165-t003:** Descriptive characteristics of the 34 included FCR PROMs (July 2011–December 2023).

Author, Year	Purpose of the Study	Language Country	Cancer Population	Name Given to the PROM in the Study	Dimensions, Assessment Method, Total Items, Name and Number of Subscales	Range of Scores and Scoring Interpretation	Response Format
1. Established PROMs with Cultural or Population-Specific Validation							
1.1 Fear of Cancer Recurrence Inventory (FCRI)—Adaptation and Validation in New Languages and Populations							
Lebel et al., 2016 [21] FCRI English	To validate the English translation of the FCRI long version	Language: English Country: Canada	*n* = 350 Breast, prostate, colorectal, and lung cancer survivors *n* = 135 at one-month retest	FCRI	Assessment Method: Self-reported and clinical interviewing The original seven-factor structure remained (42 items and 7 subscales): Triggers: 8 items Severity: 9 items Psychological Distress: 4 items Functioning Impairments: 6 items Insight: 3 items Reassurance: 3 items Coping Strategies: 9 items	Range: 0–168 42 items; 5-point Likert scale; subscale and total scores; higher scores = greater FCR Interpretation: Includes Severity subscale cut-off ≥ 13 for screening and ≥ 16 for clinical relevance	Items are rated on a 5-point scale: 0 = never/not at all 1 = rarely/a little 2 = sometimes/somewhat 3 = most of the time/a lot 4 = all the time/a great deal
Shin et al., 2017 [22] K-FCRI	To confirm the cultural equivalence, reliability, and validity of the Korean version of FCRI (K-FCRI)	Language: Korean Country: South Korea	*n* = 444 mixed cancer survivors	K-FCRI	Assessment Method: Self-report Total Items: 42 items Number of Subscales: 7 subscales Triggers: 8 items Severity: 9 items Psychological Distress: 4 items Functioning Impairments: 6 items Insight: 3 items Reassurance: 3 items Coping Strategies: 9 items	Range: 0–168 Higher scores indicate higher levels of FCR Score of ≥13 = cut off for clinically significant FCR on severity subscale	5-point Likert scale (0 = never, 4 = all the time)
Van Helmondt et al., 2017 [23] FCRI-NL	To translate and evaluate the psychometric validation of the Dutch version of the FCRI	Language: Dutch Country: Netherlands	*n* = 255 Mixed cancer survivors (88.6% female; age M = 51.0 ± 9.8)	FCRI-NL	Assessment Method: Self-report Total Items: 42 items Number of Subscales: 7 subscales Triggers: 8 items Severity: 9 items Psychological Distress: 4 items Functioning Impairments: 6 items Insight: 3 items Reassurance: 3 items Coping Strategies: 9 items	Range: 0–168 Scoring and Interpretation: Subscale and total scores are calculated by summing items (Item 13 reverse-coded). Severity subscale (FCRI-SF-NL) can be used as a standalone screener: Cut-off ≥ 13 = clinically significant FCR (high sensitivity) Cut-off ≥ 16 = clinical vs. non-clinical levels (higher specificity) Higher scores indicate greater FCR.	5-point Likert scale (0 = never, 4 = all the time)
Hovdenak Jakobsen et al., 2018 [24] The Danish version of the FCRI	To translate and pilot test the FCRI in a gynecological cancer population and validate the translated Danish version of the FCRI in a population of colorectal cancer patients	Language: Danish Country: Denmark	Pilot: *n* = 24 (endometrial) Validation: *n* = 69 (colorectal; *n* = 49 at retest)	The Danish version of the FCRI	Unidimensional Assessment Method: Self-report Total Items: 42 items Subscales: 7 Triggers: 8 items Severity: 9 items Psychological Distress: 4 items Functioning Impairments: 6 items Insight: 3 items Reassurance: 3 items Coping Strategies: 9 items	Range: 0–168 42 items; 5-point Likert scale; subscale and total scores; higher scores = greater FCR Interpretation: Includes Severity subscale cut-off ≥ 13 for screening and ≥ 16 for clinical relevance	Items are rated on a 5-point scale: 0 = never/not at all 1 = rarely/a little 2 = sometimes/somewhat 3 = most of the time/a lot 4 = all the time/a great deal
Liu et al., 2020 [25] FCRI Singapore cancer survivors	To translate and evaluate the psychometric validation of the FCRI in Mandarin in an Asian cancer population	Language: Mandarin and English Country: Singapore	*n* = 331 (Mandarin = 109; English = 222) mixed cancer survivors *n* = 219 (Mandarin = 109; English = 110) mixed cancer survivors	FCRI in Singapore cancer survivors	Assessment Method: Self-report Total Items: 42 items Number of Subscales: 7 subscales Triggers: 8 items Severity: 9 items Psychological Distress: 4 items Functioning Impairments: 6 items Insight: 3 items Reassurance: 3 items Coping Strategies: 9 items	Range: 0–168 Higher scores indicate higher levels of FCR Cut-off/Interpretation: Not explicitly defined in this study	5-point Likert scale (0 = never, 4 = all the time)
Xu et al., 2021 [19] FCRI-C	To translate and validate the FCRI in Chinese (FCRI-C), and create and validate a short form development of the FCRI-C (short FCRI-C reported in Table 3 and Table 4: 2.1)	Language: Chinese Country: China	*n* = 326 Chinese follicular lymphoma survivors	FCRI-C Long version	Assessment Method: Self-report FCRI-C Long version The original seven-factor structure remained (42 items and 7 subscales)	Range: 0–168 42 items; 5-point Likert scale; subscale and total scores; higher scores = greater FCR Cut-off scores for identifying potential clinical FCR cases were calculated using ROC-based: Full form cut-off = 83	Items are rated on a 5-point scale: 0 = never/not at all 1 = rarely/a little 2 = sometimes/somewhat 3 = most of the time/a lot 4 = all the time/a great deal
**1.2 Concerns About Recurrence Scale (CARS)—Adaptation and Validation in a New Language and Population**							
Momino et al., 2014 [26] CARS-J	To translate and validate the Japanese version of the CARS (CARS-J) among breast cancer survivors	Language: Japanese Country: Japan	*n* = 375 female breast cancer survivors	CARS-J	Dimension Assessed: Fear of cancer recurrence—multidimensional Assessment Method: Self-report No. of Items: 26 items analyzed from original 30 Subscales: Health and Death Worries (13 items) Womanhood Worries (6 items) Self-Valued Worries (5 items) Role Worries (2 items)	Score Range and Interpretation: Part 1: 4–128; Part 2: 0–124 Total range for full CARS-J: 4 to 128 Higher scores indicate greater fear of recurrence	Part 1: Overall fear of recurrence assessed using 4 items on a 6-point Likert scale (1 = not at all, 6 = continuously/terribly) Part 2: 26 items across 4 domains scored on a 5-point Likert scale (0 = not at all, 4 = extremely)
**2. FCRI-Derived Short Forms—Development, Validation, and Cross-Cultural Adaptation**							
**2.1 FCRI (Simard and Savard, 2009)** [5]**—Shortened and Adapted Versions**							
Costa et al., 2016 [27] Shortened form of FCRI 16 items	To evaluate the psychometric properties of the full 42-item FCRI using item response theory and to propose a shortened version based on item-level discrimination	Language: English Country: Australia	*n* = 286 adult melanoma survivors (moderate to high recurrence risk)	Shortened form of FCRI	Unidimensional Subscales (based on retained items): Items drawn from all seven original subscales; not restructured into new domains No. of Items: 16 Number of Subscales: 7 Triggers: 3; Severity: 2; Psychological Distress: 2; Functioning Impairments: 3; Insight: 2; Reassurance: 1; Coping; Strategies: 3 **Note:** Subscale structure derived from original FCRI; not independently validated in the 16-item form. **Note from author**: A short-form based on all seven domains is not conceptually equivalent to the Severity subscale and should not be viewed as a replacement for the Severity subscale as a screening tool. Instead, its purpose is to capture the same information as the complete FCRI, but with reduced respondent burden.	Total score range: 0–64 Based on 5-point scale across 16 items Higher scores indicate greater fear of cancer recurrence	5-point Likert scale (0 = never, 4 = all the time)
Eyrenci and Sertal Berk, 2018 [28] FCRI-24 Shortened version	To translate, culturally adapt, and evaluate the psychometric validation of the Turkish version of the FCRI	Language: Turkish Country: Turkey	*n* = 219 mixed cancer survivors with breast cancer subgroup	FCRI-24	Assessment Method: Self-report Total Items: 24 items (following EFA-based item reduction) Number of Subscales: 5 subscales Triggers: 7 items; Functioning Impairments: 2 items; Recurrence Related Meta-Cognitions: 4 items; Emotion-Focused Coping Strategies: 5 items; Quality of Life: 6 items	Score range: 0–96 Higher scores indicate higher levels of FCR Range: 0–96 Higher scores indicate higher levels of FCR	5-point Likert scale (0 = never, 4 = all the time)
Xu et al., 2021 [19] FCRI-C Shortened version 10 items	To translate the FCRI in Chinese (FCRI-C), validate, and develop a short form of the FCRI-C Short version	Language: Chinese Country: China	*n* = 326 Chinese follicular lymphoma survivors	FCRI-C Short version 10 items	FCRI-C Short version derived through IRT analysis: 10 items and four subscales Severity: 2 items; Psychological distress: 3 items; Functioning impairments: 4 items; Insight: 1 item; 10 items derived from the original 42 items with items: 9, 12, 18, 19, 20, 22, 23, 26, 27, and 28	Score Range and Interpretation: Score range: 0–40 Cut-off scores for identifying potential clinical FCR cases were calculated using ROC based: Short form cut-off = 20	Items are rated on a 5-point scale: 0 = never/not at all 1 = rarely/a little 2 = sometimes/somewhat 3 = most of the time/a lot 4 = all the time/a great deal
**2.2 FCRI Short Form (FCRI-SF) (Simard and Savard, 2009)** [5]**—Screening Variants and cut-off versions**							
Simard and Savard, 2015 [29] FCRI-SF French	To assess the capacity of the FCRI-severity subscale, to consider as a shorter form of the FCRI (FCRI-SF), to screen for clinical levels of FCR	Language: French Country: Canada	*n* = 60 French-Canadian cancer survivors, 38% breast cancer, 38% prostate cancer, 17% colorectal cancer, and 7% lung cancer. Sample; 43% female, mean age 60.3 years.	FCRI-SF	Unidimensional Assessment Method: Self-report Severity Subscale: 9 items	Score range: 0–36 Higher scores indicate higher levels of FCR A score of ≥ 13 = cut off for clinically significant FCR	9 items from the Severity subscale; rated on a 5-point Likert scale (0 = never, 4 = all the time); items reflect frequency and intensity of intrusive FCR-related thoughts
Fardell et al., 2018 [30] FCR-SF English	To evaluate the FCRI-Short Form (FCRI-SF) clinical cut-off in 2 samples	Language: English Country: Australia and Canada	Study 1 Australian Population: 167 Breast, colorectal, and melanoma cancer survivors Study 2 Canadian Population: 40 Breast, prostate, lung, and colorectal cancer survivors	FCR-SF	Unidimensional Assessment Method: Self-report, semi-structured clinical interview Severity Subscale: 9 items	Score range: 0–36 Higher scores indicate higher levels of FCR A score of ≥22 = cut off for clinically significant FCR	9 items from the Severity subscale; rated on a 5-point Likert scale (0 = never, 4 = all the time); items reflect frequency and intensity of intrusive FCR-related thoughts
Peng et al., 2019 [31] Chinese FCR-SF	To translate the FCRI-SF in Chinese and validate its clinical cut-off in Chinese breast cancer survivors	Language: Chinese Country: China	*n* = 207 breast cancer survivors, age range 19–60, stage 0–III, post-treatment	Chinese FCR-SF	Structural validity of translated Chinese version not assessed; original FCRI-SF unidimensional structure assumed Assessment Method: Self-report Severity Subscale: 9 items	Score range: 0–36 From ROC analysis, assumes a cut-off ≥ 12 as optimal for detecting high FCR (Sensitivity = 98.6%, Specificity = 35%, AUC = 0.83)	9 items from the Severity subscale; rated on a 5-point Likert scale (0 = never, 4 = all the time); items reflect frequency and intensity of intrusive FCR-related thoughts
Decat Bergerot et al., 2023 [20] FCRI-SF in Portuguese	To translate and validate the FCRI-SF and, but reported above in 3.2, the FCR4/7 scales into Portuguese	Language: Portuguese Country: Brazil	Patients with localized breast cancer (*n* = 100) and metastatic heterogeneous cancer (*n* = 100)	FCRI-SF	The FCRI-SF includes severity of a symptom, coping, functioning impairments, triggers, insight, duration, and reassurance	Range of Scores: 0 to 36 Scoring Interpretation: Item 5 is reverse-coded. The total score ranges from 0 to 36, with higher scores indicating greater severity of fear of cancer recurrence. Cut-off scores are 13 (moderate FCR) and 22 (severe FCR).	9 items. Six items use a 5-point Likert scale ranging from 0 (“not at all or never or I don’t think about it”) to 4 (“a great deal or several times a day or several hours or several years”). Three items use a Likert scale with different specific labels, ranging from 0 (“never or I don’t think about it”) to 4 (“several”).
**2.3 Fear of Progression Questionnaire Short Form (FoP-Q-SF)** [32] **and FoP-Q** [33]**—Adaptations and Validations in New Languages and Populations**							
Mahendran et al., 2020 [34] FoP-Q-SF Simplified Mandarin	To validate the English and Mandarin versions of the FoP-Q-SF	Languages: Simplified Mandarin and English Country: Singapore	*n* = 341 mixed cancer survivors	FoP-Q-SF	Unidimensional FoP-Q-Short Form 12 items	12 items; Range: 12–60 Higher scores indicate a higher level of fear of progression. Interpretation: A score of 34 or above indicates a dysfunctional level of FoP.	Scale: 5-point Likert (1 = “never” to 5 = “very often”)
Abd Hamid et al., 2021 [35] FoP-SF-M Malay	To validate the Malay version of the FoP-Q-SF (FoP-SF-M)	Language: Malay Country: Malaysia	*n* = 200 mixed cancer patients	FoP-SF-M	Unidimensional 12 items	12 items; Range: 12–60 Higher scores indicate a higher level of fear of progression. Interpretation: A score of 34 or above indicates a dysfunctional level of FoP.	Scale: 5-point Likert (1 = “never” to 5 = “very often”)
Youssef et al. 2021 [36] FoP-Q-RS German	To develop and validate the fear of Progression Questionnaire Rapid Screener (FoP-Q-RS) from the FoP-Q-SF for oncology settings	Language: German Country: Germany	*n* = 1002 mixed cancer types (registry-based sample in Leipzig)	FoP-Q-RS German	Unidimensional CFA, 5 items, no subscales	5 items; Range: 5 to 25 Higher scores indicating a higher level of FoP. A cut-off score of 12 yielded a sensitivity (SEN) of 72% (95% CI: 63–82%) and a specificity (SPE) of 70% (95% CI: 67–73%).	Scale: 5-point Likert (1 = “never” to 5 = “very often”)
Cheng et al., 2022 [37] FoP-Q-SF Traditional Chinese	To validate the Traditional Chinese version of the FoP-Q-SF	Language: Traditional Chinese Country: Hong Kong	*n* = 311 mixed cancer survivors	FoP-Q-SF Chinese	Unidimensional 12 items	12 items; Range: 12–60 Higher scores indicate a higher level of fear of progression. Interpretation: A score of 34 or above indicates a dysfunctional level of FoP.	Scale: 5-point Likert (1 = “never” to 5 = “very often”)
Silva et al., 2022 [38] FoP-Q-SF Portuguese	To translate and validate the Portuguese version of the FoP-Q-SF (short form) in cancer survivors	Language: Portuguese Country: Portugal	*n* = 220 volunteers recruited online from mixed cancer types	FoP-Q-SF Portuguese	Unidimensional 12 items belonging to four of the five subscales (excluding coping with anxiety) from the long version of FoP-Q [33]. Six items taken from the “Affective” subscale, two items from the “Occupation” subscale, two items from the “Relationship and Family” subscale and two items from the subscale “Loss of Autonomy”.	12 items; Range: 12–60 Higher scores indicate a higher level of fear of progression. 12 items summed up in four subscales: affective reactions, partnership/family issues, occupation, and loss of autonomy) with higher scores indicating a higher level of FoP.	Scale: 5-point Likert (1 = “never” to 5 = “very often”)
Hasannezhad Reskati et al., 2023 [39] FoP-Q Persian	To translate and validate the Persian version of the FoP-Q (full version) in Gastrointestinal (GI) cancer patients	Language: Persian Country: Iran	*n* = 430 GI cancer patients, aged 19–78	Persian Version FoP-Q (full version)	Multidimensional 43 items 5-factor solution via EFA (variance explained = 37%)	Scoring: Total score range not explicitly reported; subscales based on 5 factors (emotional response, employment, loss of independence, economy/family, coping)	Scale: 5-point Likert (1 = “never” to 5 = “very often”)
**3. New Developed FCR PROMs**							
**3.1 Cancer Worry Scale (CWS): Validation and Adaptation of the New PROM in New Languages and Populations**							
Custers et al., 2014 [40] CWS-8 items	To validate a Dutch version of the 8-item CWS	Language: Dutch Country: Netherlands	*n* = 194 female breast cancer patients	CWS-8	Single factor structure. Self-report. 8 items; no subscale	Range: 8–32 Higher scores indicate more frequent worries about cancer Cut-off score for detecting severe levels of FCR: Low: ≤ 13, High: ≥14 Cut-off score for screening: Low: ≤11, High: ≥12	Items are rated on a 4-point Likert scale ranging from “1 = never” to “4 = almost always.”
Custers et al., 2018 [41] CWS-6 items	To validate a Dutch shorter 6-item version of the CWS and assess cut-off scores for clinical levels of FCR	Language: Dutch Country: Netherlands	*n* = 981 cancer survivors with breast, prostate, colorectal, and GIST cancers	CWS-6	Single factor structure. Self-report. 6 items; no subscale	Range: 6–24 Higher scores indicate more frequent worries about cancer Cut-off score for detecting severe levels of FCR: Low ≤ 11, High: ≥12 Cut-off score for screening: Low: ≤9, High: ≥10	Items are rated on a 4-point Likert scale ranging from “1 = never” to “4 = almost always.”
Chirico et al., 2022 [42] CWS-8 items Italian	To validate an Italian version of the 8-item CWS	Italian	108 breast cancer survivors	Italian version-CWS	8 items and 2 factors: Cancer worries: 4 items Cancer worry impact: 4 items	Range: 8–32 Higher scores indicate more frequent worries about cancer Cut-off score for differentiating fearful from non-fearful: Non-fearful: ≤15, Fearful: ≥16	Items are rated on a 4-point Likert scale ranging from “1 = never” to “4 = almost always.”
**3.2 FCR4 and FCR7 (Humphris et al., 2018)** [43]**: Validation and Adaptation of the New PROM in New Languages and Populations**							
Humphris et al., 2018 [43] FCR4/FCR7	To validate two versions of FCR screening with the FCR7 and the short form FCR4	Language: English Country: UK	Total N= 259 206 breast cancer survivors and 53 colorectal cancer survivors	FCR4 FCR7	Both tools are unidimensional FCR-7: 7 items FCR-4: 4 items FCR7 is the FCR4 with 3 additional items. The first four items feature anxiety, worry, and feelings with the return of disease. Item 6 is a behavioral response to FCR.	FCR–4 Range: 4 to 20 FCR–7 Range: 6 to 40 Interpretation: Higher scores indicate greater FCR. Cut-off Scores: 60th percentile (moderate): FCR4 ≥ 10; FCR7 ≥ 17 90th percentile (high/clinical concern): FCR4 ≥ 15; FCR7 ≥ 27 These cut-offs were pragmatically derived in consultation with clinicians, as stated by the authors.	FCR4 (Items 1–4) and FCR 7 (Items 1–7): Items 1 to 6 are rated on a 5-point Likert scale: 1 = not at all, 2 = occasionally, 3 = sometimes, 4 = most of the time, 5 = all the time Last item 7 rated on a 10-point Likert scale: 0 = Not at all to 10 = A great deal
Yang et al., 2019 [44] FCR-7 Simplified Chinese	To translate the FCR-7 into Chinese and validate the scale with Chinese cancer patients	Language: Simplified Chinese, Mandarin and Cantonese Country: China	Total *n* = 1025 mixed cancer patients (90% female) with specific: 803 breasts 109 lungs 84 colon-rectum 29 nasopharynx	Chinese version of the FCR-7	Unidimensional FCR scale 7 items	FCR–4 Range: 4 to 20 FCR–7 Range: 6 to 40 Interpretation: Higher scores indicate greater FCR. No cut-off has been reported other than the statistical 60th (score 17) and 90th (score 27) percentiles, which have been regarded as levels for “moderate” and “high” reports of patients’ FCR, respectively.	Items 1–6 rated on a 5-point Likert with 1 = Not at all to 5 = All the time. Item 7: 0–10 scale (0 = Not at all, 10 = A great deal).
Lee et al., 2020 [45] FCR7 Traditional Chinese	To validate the FCR7 in Chinese for lung cancer patients	Language: Traditional Chinese, Mandarin Country: Taiwan	Patients with lung cancer (*n* = 160)	FCR7-C	Unidimensional FCR scale: 7 items The first 4 items measure level of concern about cancer recurrence, the next two determine extent of FCR affecting daily life, and the last item determines how fear intrudes thoughts and activities	FCR–7 Range: 6 to 40 Interpretation: Higher scores indicate greater FCR. Although no definitive cut-off exists, the total or single-item score may be used to preliminarily identify FCR severity, as suggested by the authors.	Items 1–6 rated on a 5-point Likert with 1= Not at all to 5 = All the time. Item 7: 0–10 scale (0 = Not at all, 10 = A great deal).
Braun et al., 2022 [46] FCR6-Brain	To validate the FRC 6–Brian, in patients with primary brain tumor (PBT) and their caregivers	Language: English Country: USA	Patients with primary brain tumor (*n* = 165) and caregivers (but excluded in this SR); *n* = 117)	FCR6-Brain	Unidimensional single-factor structure Self-report 6 items No subscales	FCR6-Brain Range: 5–35 Higher scores indicate higher FC. Severity guidelines: Scores of ≥18 (60th percentile) is considered clinically subthreshold. A score ≥31 (90th percentile) is to be flagged as clinically significant FCR.	The FCR6-Brain with items 1 to 5: rated on a 5-point Likert scale from “not at all” to “all the time.” Item 6 is rated on a 10-point scale from “0 = not at all” to “10 = all the time”
Iglesias-Puzas et al., 2022 [47] FCR-7 Spanish	To validate the FCR7 in Spanish for patients with non-metastatic melanoma	Language: Spanish Country: Spain	Patients with non-metastatic melanoma (*n* = 123)	Spanish version of the FCR-7	Unidimensional FCR scale: 7 items The first 4 items measure level of concern about cancer recurrence, the next two determine extent of FCR affecting daily life, and the last item determines how fear intrudes thoughts and activities	FCR–7 Range: 6 to 40 Interpretation: Higher scores indicate greater FCR.	Items 1–6 rated on a 5-point Likert with 1 = Not at all to 5 = All the time. Item 7: 0–10 scale (0 = Not at all, 10 = A great deal).
Nandakumar et al., 2022 [48] FCR-7-T in Tamil	To translate and validate the FCR7 scale into the regional language Tamil among breast cancer survivors	Language: Tamil Country: India	106 female breast cancer survivors	FCR7-Tamil (FCR7-T)	Unidimensional FCR scale: 7 items No subscales Assessed via total score only	Total score ranges from 6 to 40; higher scores indicate greater FCR.	Items 1–6 rated on a 5-point Likert with 1 = Not at all to 5 = All the time. Item 7: 0–10 scale (0 = Not at all, 10 = A great deal).
Bergerot et al., 2023 [20] FCR4/7 Portuguese	To translate and validate the FCR4/7 and, but reported above in 2.2, the FCRI-SF scales into Portuguese	Language: Portuguese Country: Brazil	Patients with localized breast cancer (*n* = 100) and metastatic heterogeneous cancer (*n* = 100)	FCR4/7	The FCR4 assesses symptoms of anxiety and worry concerning recurrence, while the FCR7 extends upon these items to include a cognitive processing component and behavioral responses	FCR–7 Range: 6 to 40 Interpretation: Higher scores indicate greater FCR. A cut-off score of 17 or above is considered moderate FCR, and 27 or above, severe FCR.	Items 1–6 rated on a 5-point Likert with 1 = Not at all to 5 = All the time. Item 7: 0–10 scale (0 = Not at all, 10 = A great deal).
**3.3 Concerns About Recurrence Questionnaire (CARQ) CARQ-4 (Thewes et al., 2015)** [49]**: Validation of the New PROM in a New Language and Populations**							
Thewes et al., 2015 [49] CARQ-4	To describe the development of a new measure of FCR for breast cancer survivors, the Concerns about Recurrence Questionnaire (CARQ), and to report on its initial validation in Australian and Danish samples.	Language: English and Danish Country: Australia and Denmark	Australian sample: *n* = 218, women early-stage breast cancer (Stages 0–2) Danish sample: *n* = 2001, women with breast cancer (Stages 1–3)	CARQ-4 (Concerns About Recurrence Questionnaire—4 items version)	Unidimensional Assessment Method: Self-report Total Items: 4 No subscales (all items assess core elements of FCR: frequency, intrusiveness, distress, and perceived risk)	Total score range: 0–40 Higher scores indicate greater fear of cancer recurrence Clinical cut-offs of ≥12 for the CARQ-4 and ≥10 for the CARQ-3 to distinguish high from low levels of FCR.	Items 1–3: 11-point Likert scale (0 = not at all to 10 = a great deal) Item 4 (transformed perceived risk): originally 0–100%, transformed to 0–10 scale Final total score = sum of items (range 0–40)
**3.4 FCR-1 (Rudy et al., 2020)** [50]**—Validation of a New PROM, and Adaptations in New Languages and Populations**							
Rudy et al., 2020 [50] FCR-1	To develop and validate the first one-item measure of FCR	Language: English Country: Canada	69 women with breast cancer and gynecological cancer	FCR-1	Unidimensional FCR scale 1 item Available as a 0 to 100 scale or 0 to 10 scale depending on patient preference of format.	Total score range 0–100 or 0 –10, depending on patient preference of format Higher score indicates higher FCR Clinical cut-offs of ≥45 for the 0–100 scale and ≥4.5 for the 0–10 scale	Item is rated on a visual scale from 0–100 or 0–10
Smith et al., 2023 [51] FCR-1r	To evaluate the validity of a one-item FCR (FCR-1r) and determine its screening performance	Language: English Country: Australia	107 mixed cancer survivors from two studies. Study 1: *n* = 54 Study 2: *n* = 53	FCR-1r	Unidimensional FCR scale 1 item Available as a 0–10 visual analog scale Embedded in the ESAS-r in Study 1 (with standard symptom items); standalone in Study 2 (collected as part of routine care along with the FCRI-SF)	Total score range 0–10 Higher scores indicate higher FCR Clinical cut-offs of ≥5	Item is rated on a visual scale from 0–10
Lyhne et al., 2023 [52] FCR-1r Danish	To translate and validate the FCR-1r into Danish among long-term colorectal cancer survivors and assess its screening capacity in older and younger survivors	Language: Danish Country: Denmark	*n* = 1654 colorectal cancer survivors	FCR-1r Danish	Unidimensional FCR scale 1 item Available as a 0–10 visual analog scale distributed from November 2021 to May 2023. Recruited through the Danish Clinical Quality Program-National Clinical Registries using both paper-and-pencil questionnaires.	Total score range 0–10 Higher scores indicate higher FCR Clinical cut-offs of ≥5	Item is rated on a visual scale from 0–10

Abbreviations: FCRI = Fear of Cancer Recurrence Inventory; FoP-Q = Fear of Progression Questionnaire; CWS = Cancer Worry Scale; CARS = Concerns About Recurrence Scale; CARQ = Concerns About Recurrence Questionnaire. Note: Numbering in this table (e.g., 1.1, 2.1, 2.2) refers to PROM categories and does not correspond to manuscript section numbering.

**Table 4 healthcare-13-02165-t004:** Summary of overall measurement property ratings and quality of evidence per PROM.

PROMs	Structural Validity (mp1)		Internal Consistency (mp2)		Reliability (mp3)		Measurement Error (mp4)		Hypothesis Testing (mp5)		Cross-Cultural Validity (mp6)		Criterion Validity (mp7)		Responsiveness (mp8)		Measure Category as per COSMIN
1. Established PROMs with Cultural or Population-Specific Validation																	
1.1 Fear of Cancer Recurrence Inventory (FCRI) (Simard & Savard, 2009) [5]—Adaptation and Validation in New Languages and Populations																	
	mp1		mp2		mp3		mp4		mp5		mp6		mp7		mp8		Category
Lebel et al., 2016 [21] FCRI English	+	High	+	High	+	High	?	Very Low	+	High	?	Very Low	+	Moderate	?	Very Low	A
Shin et al., 2017 [22] K-FCRI Korean	+	High	+	High	+	High	?	Very Low	+	High	+	High	+	Moderate	?	Very Low	A
van Helmondt et al., 2017 [23] FCRI-NL Dutch	+	High	+	High	+	High	?	Very Low	+	High	?	Very Low	+	Moderate	?	Very Low	A
Hovdenak Jakobsen et al., 2018 [24] FCRI Danish	?	Very Low	?	Very Low	+	High	?	Very Low	+	High	?	Very Low	?	Very Low	+	High	B
Liu et al., 2020 [25] FCRI Mandarin	+	High	+	High	+	High	?	Very Low	+	High	+	High	+	Moderate	?	Very Low	A
Xu et al., 2021 [19] FCRI-C Chinese long version	+	High	+	High	+	High	?	Very Low	+	High	−	Moderate	+	High	?	Very Low	A
**1.2 Concerns About Recurrence Scale (CARS)—Adaptation and Validation in a New Language and Population**																	
Momino et al., 2014 [26] CARS-J	+	Moderate	+	High	?	Very Low	?	Very Low	+	High	?	Very Low	+	Moderate	?	Very Low	A
**2. FCRI-Derived Short Forms—Development, Validation, and Cross-Cultural Adaptation**																	
**2.1 FCRI (Simard and Savard, 2009)** [5]**—Shortened and Adapted Versions**																	
	**mp1**		**mp2**		**mp3**		**mp4**		**mp5**		**mp6**		**mp7**		**mp8**		**Category**
Costa et al., 2016 [27] FCRI Shorter version	+	High	?	Very Low	?	Very Low	?	Very Low	?	Very Low	?	Very Low	?	Very Low	?	Very Low	B
Eyrenci and Sertel Berk, 2018 [28] FCRI Turkish shorter version	+	Moderate	+	High	?	Very Low	?	Very Low	+	High	?	Very Low	+	Moderate	+	Very Low	A
Xu et al., 2021 [19] FCRI-C Chinese shorter version	+	High	+	High	+	High	?	Very low	+	High	−	Moderate	+	High	?	Very Low	A
**2.2 FCRI Short Form (FCRI-SF)** [29]**—Screening Variants and Cut-Off Versions**																	
Simard and Savard, 2015 [29] FCRI-SF French	+	Moderate	+	Moderate	?	Very Low	?	Very Low	+	High	?	Very Low	+	High	?	Very Low	A
Fardell et al., 2018 [30] FCRI-SF	NA	NA	NA	NA	?	Very Low	?	Very Low	+	High	?	Very Low	+	High	?	Low	A
Peng et al., 2019 [31] FCRI-SF Chinese	?	Very Low	?	Very Low	?	Very Low	?	Very Low	?	Low	?	Very Low	+	High	?	Very Low	C
Decat Bergerot et al., 2023 [20]	+	Moderate	+	High	?	Very Low	?	Very Low	?	Very Low	?	Very Low	?	Very Low	?	Very Low	A
**2.3 Fear of Progression Questionnaire Short Form (FoP-Q-SF)** [32] **and FoP-Q** [33]**—Adaptations and Validations in New Languages and Populations**																	
Mahendran et al., 2020 [34] FoP-Q-SF Simplified Mandarin	+	High	+	High	+	High	?	Very Low	+	High	?	Very Low	+	Moderate	?	Very Low	A
Abd Hamid et al., 2021 [35] FoP-Q-SF-M Malay	+	High	+	High	?	Very Low	?	Very Low	+	High	?	Very Low	?	Very Low	?	Very Low	A
Youssef et a., 2021 [36] FoP-Q-RS German	+	High	+	Moderate	?	Very Low	?	Very Low	+	Moderate	?	Very Low	+	High	?	Very Low	A
Cheng et al., 2022 [37] FoP-Q-SF Traditional Chinese	+	High	+	High	?	Very Low	?	Very Low	+	Moderate	?	Very Low	?	Very Low	?	Very Low	A
Silva et al., 2022 [38] FoP-Q-SF Portuguese	+	Moderate	+	High	?	Very Low	?	Very Low	+	Moderate	?	Very Low	−	Moderate	?	Very Low	A
Hasannezhad Reskati et al., 2023 [39] FoP-Q Persian	−	Moderate	?	High	+	Moderate	?	Very Low	+	High	?	Very Low	?	Very Low	?	Very Low	B
**3. New Developed FCR PROMs**																	
**3.1 Cancer Worry Scale (CWS) (Custers et al., 2014)** [40]**: Validation and Adaptations of the New PROM in New Languages and Populations**																	
	**mp1**		**mp2**		**mp3**		**mp4**		**mp5**		**mp6**		**mp7**		**mp8**		**Category**
Custers et al., 2014 [40] CWS-8 items	+	High	+	High	?	Very low	?	Very Low	+	High	?	Very Low	+	High	?	Very Low	A
Custers et al., 2018 [41] CWS-6 items	+	High	+	High	?	Very low	?	Very Low	+	High	?	Very Low	+	High	?	Very Low	A
Chirico et al., 2022 [42] CWS-8 items	+	Moderate	+	High	+	Moderate	?	Very Low	+	High	?	Very Low	+	High	?	Very Low	A
**3.2 FCR4 and FCR7 (Humphris et al., 2018)** [43]**: Validation and Adaptations of the New PROM in New Languages and Populations**																	
Humphris et al., 2018 [43] FCR4 and FCR 7	+	Moderate	+	High	?	Very low	?	Very Low	+	High	?	Very Low	+	High	?	Very Low	A
Yang et al., 2019 [44] FCR-7 Chinese	+	High	+	High	+	High	?	Very Low	+	High	?	Very Low	+	Moderate	?	Very Low	A
Lee et al., 2020 [45] FCR-7 Chinese	+	High	+	High	?	Very low	?	Very low	+	High	?	Very Low	+	Moderate	?	Very Low	A
Braun et al., 2022 [46] FCR6-Brain	+	Moderate	+	High	?	Very low	?	Very Low	+	High	?	Very Low	+	Moderate	?	Very Low	A
Iglesias-Puzas et al., 2022 [47] FCR-7 Spanish	?	Low	?	High	+	Moderate	?	Very Low	+	Low	?	Very Low	?	Very Low	?	Very Low	B
Nandakumar et al. 2022 [48] FCR7 Tamil	?	Very Low	?	Very Low	+	Moderate	?	Very Low	+	High	?	Very Low	+	Moderate	?	Very Low	B
Bergerot et al., 2023 [20] FCR4/7 Portuguese	+	Moderate	+	High	?	Very Low	?	Very Low	?	Very Low	?	Very Low	?	Very Low	?	Very Low	A
**3.3 Concerns About Recurrence Questionnaire (CARQ) CARQ-4 (Thewes et al., 2015)** [49]**: Validation of a New PROM**																	
Thewes et al., 2015 [49] CARQ-4	+	High	+	High	+	High	?	Very Low	+	High	−	Moderate	+	High	?	Very Low	A
**3.4 FCR-1 (Rudy et al., 2020)** [50]**—Validation of a New PROM, and Adaptations in New Languages and Populations**																	
Content Validity **^1^**			mp2 NA **^1^**														
Rudy et al., 2020 [50] FCR-1	+	High	NA		+	High	?	Very Low	+	High	?	Very Low	+	High	+	High	A
Smith et al., 2023 [51] FCR-1r	+	Low	NA		+	Low	?	Very Low	+	Moderate	?	Very Low	+	Moderate	?	Very Low	A
Lyhne et al., 2023 [52] FCR-1r Danish	+	High	NA		+	High	?	Very Low	+	Moderate	?	Very Low	+	High	?	Very Low	A

Notes: Each measurement property is presented with their measurement properties ratings followed by their quality of evidence (QE), where Measurement properties rating refers to the COSMIN assessment rated against criteria for good measurement properties, where each measurement property gets rated as either sufficient (+), insufficient (−), or indeterminate (?). QE is rated as follows: (1) high, (2) moderate, (3) low, and (4) very low. These ratings are derived using the Modified GRADE approach, assessing measurement quality against risk of bias and determining if downgrades are necessary. In cases where the criteria are not applicable (NA) according to COSMIN, the notion NA is marked. QE is determined by combining the measurement property rating (e.g., +, ? or −) with its measurement quality assessment (e.g., very good, adequate, doubtful, inadequate) per the GRADE approach. The COSMIN Measure Category summarizes the PROM’s overall classification (e.g., Category A, B, C). Category A = strongest evidence; Category B = acceptable with caution; Category C = not recommended until further validation. ^1^ For single-item instruments, COSMIN does not require evidence of structural validity; instead, sufficient content (face) validity and any evidence of reliability (e.g., internal consistency, criterion validity) are acceptable for Category A assignment. Numbering in this table (e.g., 1.1, 2.1, 2.2) refers to PROM categories and does not correspond to manuscript section numbering.

## Data Availability

All data has been made available in Appendix A, and Table 3 and Table 4.

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
