# Peer review of "Systematic Review of Fear of Cancer Recurrence Patient-Reported Outcome Measures: Evaluating Methodological Quality and Measurement Properties Using the COSMIN Checklist"

_healthcare, 2025, doi:10.3390/healthcare13172165_

Round 1

Reviewer 1 Report

Comments and Suggestions for Authors

The manuscript addresses an important and clinically relevant topic. The study is methodologically rigorous, following a full ten-step COSMIN framework and PRISMA guidelines. It provides a structured and evidence-based synthesis of 34 PROMs from 32 studies across diverse cultural and clinical contexts. The results are highly relevant for both clinical screening and research purposes, and the discussion appropriately contextualizes the findings within the current landscape of survivorship care.

However, there are still some issues that require clarification before publication:

  1. While the manuscript outlines the use of the PICO framework, some of the population inclusion/exclusion criteria (for genetic risk carriers or those undergoing screening) would benefit from clearer justification in the methods section. Please also specify if any pediatric or AYA populations were deliberately excluded or absent.
  2. The narrative and summary tables are comprehensive, but the manuscript would benefit from an appendix table that lists each PROM’s COSMIN rating across all domains alongside the rationale for each. This would help readers compare tools more efficiently.
  3. Several long and dense sentences (especially in the Introduction and Methods) may reduce readability. Consider breaking down some complex passages for better clarity and flow. A professional language edit is recommended to enhance precision and readability across sections.
  4. While the manuscript robustly addresses methodological quality, it would be helpful to provide a more explicit practical guide for clinicians.
  5. Many PROMs had insufficient or indeterminate cross-cultural validity (often due to missing DIF or MG-CFA analyses). Consider adding a stronger commentary on how this limits global generalizability and what future validation work is needed.
  6. The manuscript references PROSPERO registration (CRD42023453783), but it would be prudent to explicitly confirm whether any protocol amendments were made after registration and if the full protocol is publicly available.
  7. Some tables (Table 4 with COSMIN ratings) are highly detailed but dense. Consider using color shading or typographic emphasis (bolding/highlighting) to help readers identify Category A/B/C instruments briefly.
Comments on the Quality of English Language

Several long and dense sentences (especially in the Introduction and Methods) may reduce readability. Consider breaking down some complex passages for better clarity and flow. A professional language edit is recommended to enhance precision and readability across sections.

Author Response

Dear Reviewer 1,

Comment: The manuscript addresses an important and clinically relevant topic. The study is methodologically rigorous, following a full ten-step COSMIN framework and PRISMA guidelines. It provides a structured and evidence-based synthesis of 34 PROMs from 32 studies across diverse cultural and clinical contexts. The results are highly relevant for both clinical screening and research purposes, and the discussion appropriately contextualizes the findings within the current landscape of survivorship care.

However, there are still some issues that require clarification before publication:

  1. While the manuscript outlines the use of the PICO framework, some of the population inclusion/exclusion criteria (for genetic risk carriers or those undergoing screening) would benefit from clearer justification in the methods section. Please also specify if any pediatric or AYA populations were deliberately excluded or absent.

Response R1 comment 1 (R1C1):

Thank you for this valuable suggestion. The eligibility criteria have been clarified in the Methods section. The revised text now specifies that studies were eligible if they included adults (≥18 years) with a confirmed diagnosis of malignant cancer and were excluded if fewer than 80% of participants had such a diagnosis. Additional exclusions were applied to studies focusing on genetic risk carriers, individuals with benign tumors, general cancer screening populations, and pediatric or AYA groups. These refinements ensure that the review is restricted to PROMs validated in adult cancer populations with a confirmed malignant diagnosis.

  1. The narrative and summary tables are comprehensive, but the manuscript would benefit from an appendix table that lists each PROM’s COSMIN rating across all domains alongside the rationale for each. This would help readers compare tools more efficiently.

R1C2:

Thank you for this suggestion. An appendix table detailing each PROM’s COSMIN rating across all measurement properties, together with the rationale for each determination, has already been provided as Supplemental A. This supplemental material is referenced in the manuscript text, and it allows readers to efficiently compare tools across all domains. To improve visibility, an additional cross-reference to Supplemental A has been inserted in the Results section.

  1. Several long and dense sentences (especially in the Introduction and Methods) may reduce readability. Consider breaking down some complex passages for better clarity and flow. A professional language edit is recommended to enhance precision and readability across sections.

RIC3:

Thank you for highlighting this important point. The Introduction and Methods sections have been reviewed and revised for clarity by co-author M. Singh, who is a professional editor. In addition, as recommended and in line with the journal’s offer, the revised manuscript will be submitted to the journal’s professional language editing service once all reviewer comments have been addressed.

  1. While the manuscript robustly addresses methodological quality, it would be helpful to provide a more explicit practical guide for clinicians.

R1C4

Thank you for this constructive suggestion. To strengthen the clinical applicability of the review, additional text has been added to the Implications for Clinical Practice and Research section (Section 4.4). The new paragraph clarifies how Category A, B, and C PROMs should be interpreted for clinical use and highlights which instruments are most appropriate for brief screening versus comprehensive assessment. The revised text also directs readers to Supplemental A and B for structured comparisons to guide instrument selection in practice.

  1. Many PROMs had insufficient or indeterminate cross-cultural validity (often due to missing DIF or MG-CFA analyses). Consider adding a stronger commentary on how this limits global generalizability and what future validation work is needed.

R1C5

Thank you for this important observation. The Implications for Clinical Practice and Research section (4.4) has been revised to include a stronger commentary on cross-cultural validity. The added text highlights that many PROMs demonstrated insufficient or indeterminate cross-cultural evidence, largely due to the absence of differential item functioning (DIF) or multi-group confirmatory factor analysis (MG-CFA). This limitation constrains the global generalizability of PROMs and underscores the need for future validation studies to prioritize rigorous equivalence testing across diverse linguistic and cultural groups.

  1. The manuscript references PROSPERO registration (CRD42023453783), but it would be prudent to explicitly confirm whether any protocol amendments were made after registration and if the full protocol is publicly available.

R1C6

This review was prospectively registered with PROSPERO (CRD42023453783), and the full protocol is publicly accessible at https://www.crd.york.ac.uk/PROSPERO/view/CRD42023453783. Following protocol registration, one amendment was made: single-item FCR measures were also included, given their increasing recommendation in the literature as rapid screening options. No other changes were introduced.

  1. Some tables (Table 4 with COSMIN ratings) are highly detailed but dense. Consider using color shading or typographic emphasis (bolding/highlighting) to help readers identify Category A/B/C instruments briefly.

RIC7

Thank you for this helpful suggestion. Tables have been reformatted to improve readability. Category A/B/C designations are now visually emphasized with color shading, and a footnote has been added to clarify the meaning of each category.

  1. Comments on the Quality of English Language

Several long and dense sentences (especially in the Introduction and Methods) may reduce readability. Consider breaking down some complex passages for better clarity and flow. A professional language edit is recommended to enhance precision and readability across sections.

R1C8

Thank you for this observation. The manuscript has been revised to improve readability, with complex sentences in the Introduction and Methods broken down for clarity and flow. In addition, the revised version will be submitted to the journal’s professional language editing service, as recommended, to ensure precision and consistency throughout.

Reviewer 2 Report

Comments and Suggestions for Authors

Using the COSMIN methodology, this systematic review comprehensively examines patient-reported outcome measures (PROMs) assessing fear of cancer recurrence (FCR). Thirty-four scales developed or adapted between 2011 and 2023 were analyzed using data from 32 studies. The study aims to contribute to the identification of measures suitable for clinical or research use. Although the study is noteworthy in its subject matter, the authors must address some deficiencies before publication.

  1. Lines: 26–33: While the purpose of the article has been stated, it should be more clearly stated how this study differs from similar previous systematic reviews and what new contributions it makes to the literature.
  2. Although COSMIN procedures were used, a straightforward research question was not clearly stated. A clear hypothesis or question should be stated based on the PICO structure.
  3. Lines: 221–222: Considering all psychometric properties equally ignores that some properties (e.g., validity) are more critical than others. A weighted assessment approach may be more meaningful.
  4. Lines: 329–338: Although a systematic review was conducted, presenting the results narratively resulted in a lack of numerical summaries. A meta-analytic synthesis of at least some subgroups would benefit the article.
  5. The recommendations presented at the end of the article are quite general. Clinical practitioners should be given more specific guidance, such as which scale is appropriate for which patient group.

The authors' focus on the above points would increase both the scientific value and the applicability of the study.

Author Response

Dear Reviewer 2,

Comment: Using the COSMIN methodology, this systematic review comprehensively examines patient-reported outcome measures (PROMs) assessing fear of cancer recurrence (FCR). Thirty-four scales developed or adapted between 2011 and 2023 were analyzed using data from 32 studies. The study aims to contribute to the identification of measures suitable for clinical or research use. Although the study is noteworthy in its subject matter, the authors must address some deficiencies before publication.

  1. Lines: 26–33: While the purpose of the article has been stated, it should be more clearly stated how this study differs from similar previous systematic reviews and what new contributions it makes to the literature.

R2C1

Thank you for this valuable comment. The Introduction has been revised to more clearly articulate how this review differs from prior systematic reviews and the unique contributions it makes. A new paragraph has been added after the description of the IPOS FORwards initiative. This addition highlights that earlier reviews did not apply the full COSMIN methodology, did not integrate a GRADE approach, and did not systematically evaluate single-item measures or cross-cultural adaptations published since 2011. The revised text makes clear that this review represents the first comprehensive COSMIN-based appraisal of both multi-item and single-item PROMs, with structured comparison of translations and cultural validations.

  1. Although COSMIN procedures were used, a straightforward research question was not clearly stated. A clear hypothesis or question should be stated based on the PICO structure.

R2C2

Thank you for this suggestion. The Introduction has been revised to include a clearly stated research question framed using the PICO structure. The revised text now specifies: “The primary research question guiding this review was: Which patient-reported outcome measures assessing fear of cancer recurrence in adults with a confirmed cancer diagnosis demonstrate sufficient measurement properties, as evaluated using the COSMIN criteria?”

  1. Lines: 221–222: Considering all psychometric properties equally ignores that some properties (e.g., validity) are more critical than others. A weighted assessment approach may be more meaningful.

R2C3

Thank you for this observation. The review followed the standardized COSMIN methodology, which applies a structured approach to rating measurement properties without weighting domains differently. To clarify this point, the Methods, under step 4, have been updated to note that although certain properties (e.g., content validity, structural validity, internal consistency) are prioritized in COSMIN’s decision rules, the methodology does not support weighted scoring across all domains.

  1. Lines: 329–338: Although a systematic review was conducted, presenting the results narratively resulted in a lack of numerical summaries. A meta-analytic synthesis of at least some subgroups would benefit the article.

R2C4

Thank you for this suggestion. A quantitative meta-analysis was not feasible due to heterogeneity in study designs, populations, and psychometric outcomes across included studies. To improve clarity, additional numerical summaries have been emphasized in the Results tables and text, and Supplemental A provides detailed numerical ratings and rationales for each instrument. This approach is consistent with COSMIN recommendations for synthesizing evidence on measurement properties.

  1. The recommendations presented at the end of the article are quite general. Clinical practitioners should be given more specific guidance, such as which scale is appropriate for which patient group. The authors' focus on the above points would increase both the scientific value and the applicability of the study.

R2C5

Thank you for this helpful recommendation. The Implications for Clinical Practice and Research section (4.4) has been expanded to provide more specific guidance. The text now clarifies which PROMs are best suited for brief screening (e.g., FCR-1, FCRI-SF, FoP-Q-SF), for comprehensive assessment (e.g., FCRI long form and validated adaptations), and for use in specific contexts (e.g., caution in cross-cultural settings where equivalence has not been established). In addition, a new summary table has been added as Supplemental B, which highlights the five PROMs most strongly supported for clinical use.

Reviewer 3 Report

Comments and Suggestions for Authors

This is a high-quality systematic review that thoroughly applies the COSMIN methodology to evaluate fear of cancer recurrence (FCR) PROMs published between 2011 and 2023. The paper offers significant value to clinicians and researchers seeking validated tools for FCR assessment.

Please, consider clarifying whether and how publication bias was assessed (e.g., whether PROSPERO registration or grey literature were considered).If space allows, include a brief discussion of the potential clinical implications of using PROMs categorized as B or C in low-resource settings, where Category A tools may be inaccessible.It may be helpful to summarize in a single table the top 5 PROMs recommended for clinical screening vs. longitudinal follow-up, for rapid consultation by readers.

Author Response

Dear Reviewer 3,

Comment: This is a high-quality systematic review that thoroughly applies the COSMIN methodology to evaluate fear of cancer recurrence (FCR) PROMs published between 2011 and 2023. The paper offers significant value to clinicians and researchers seeking validated tools for FCR assessment.

  1. Please, consider clarifying whether and how publication bias was assessed (e.g., whether PROSPERO registration or grey literature were considered).
  2. If space allows, include a brief discussion of the potential clinical implications of using PROMs categorized as B or C in low-resource settings, where Category A tools may be inaccessible.
  3. It may be helpful to summarize in a single table the top 5 PROMs recommended for clinical screening vs. longitudinal follow-up, for rapid consultation by readers.

R3C1: Publication bias assessment

Thank you for raising this point. This review was prospectively registered in PROSPERO (CRD42023453783), ensuring transparency of the protocol. Grey literature (e.g., theses, non-peer-reviewed sources) was deliberately excluded, consistent with COSMIN guidelines and to ensure methodological rigor. While formal publication bias analysis (e.g., funnel plots) is not applicable to psychometric systematic reviews, registration and explicit eligibility criteria minimized bias in study selection. This clarification has been added to the Methods section.

Manuscript addition, Methods – Step 2 (Eligibility Criteria):

Grey literature (e.g., theses, non-peer-reviewed sources) was excluded to maintain methodological rigor, consistent with COSMIN recommendations. Although no formal statistical assessment of publication bias was conducted due to the heterogeneity of psychometric outcomes, prospective PROSPERO registration and transparent eligibility criteria reduced risk of bias in study selection.

R3C2: Clinical implications of using Category B or C PROMs in low-resource settings

We agree this is an important consideration. The Implications for Clinical Practice and Research section (4.4) has been revised to acknowledge that while Category A tools remain the gold standard, in low-resource or language-restricted settings, Category B tools may be cautiously used where Category A instruments are not yet available. Category C tools should generally be avoided, but may have exploratory value in generating preliminary insights where no other validated measures exist.

Manuscript addition, Section 4.4:
In low-resource or language-restricted settings, Category B PROMs may be cautiously considered when Category A tools are unavailable, provided results are interpreted conservatively. Category C PROMs should generally be avoided in clinical practice until further validation is available, though they may hold exploratory value in contexts where no alternative instruments exist.

R3C3: Summarizing top PROMs for rapid consultation
We thank the reviewer for this helpful suggestion. In response, a new summary table has been added as Supplemental B, which highlights the five PROMs most strongly supported for clinical use. To maintain clarity in the manuscript structure, this table is cross-referenced in Section 4.4 (Implications for Clinical Practice and Research), where practical recommendations for clinicians are discussed, rather than in the Results section. We felt this placement was most appropriate, as Supplemental B is intended as a pragmatic tool for clinical decision-making rather than part of the psychometric evidence synthesis (already fully presented in Tables 3–4 and Supplemental A). The revised Section 4.4 now directs readers to Supplemental B for rapid consultation on the top recommended PROMs for screening and longitudinal follow-up.

Manuscript addition, Section 4.4 4.4. Implications for Clinical Practice and Research

For rapid reference, Supplemental B summarizes the five PROMs most strongly supported for clinical use, highlighting their key features, strengths, and limitations to guide clinicians in selecting appropriate instruments for screening or longitudinal follow-up.